# Interplay between VSD, pore, and membrane lipids in electromechanical coupling in HCN channels

**Ahmad Elbahnsi[1†], John Cowgill[1†], Verena Burtscher[2], Linda Wedemann[1], Luise Zeckey[1], Baron Chanda[2], Lucie Delemotte[1]\***

[1]Department of Applied Physics of Science for Life Laboratory, KTH Royal Institute of Technology, Stockholm, Sweden; [2]Center for Investigation of Membrane Excitability Diseases, Department of Anesthesiology, Washington University School of Medicine, Saint Louis, United States

**Abstract** Hyperpolarized-activated and cyclic nucleotide-gated (HCN) channels are the only members of the voltage-gated ion channel superfamily in mammals that open upon hyperpolarization, conferring them pacemaker properties that are instrumental for rhythmic firing of cardiac and neuronal cells. Activation of their voltage-sensor domains (VSD) upon hyperpolarization occurs through a downward movement of the S4 helix bearing the gating charges, which triggers a break in the alpha-helical hydrogen bonding pattern at the level of a conserved Serine residue. Previous structural and molecular simulation studies had however failed to capture pore opening that should be triggered by VSD activation, presumably because of a low VSD/pore electromechanical coupling efficiency and the limited timescales accessible to such techniques. Here, we have used advanced modeling strategies, including enhanced sampling molecular dynamics simulations exploiting comparisons between non-domain swapped voltage-gated ion channel structures trapped in closed and open states to trigger pore gating and characterize electromechanical coupling in HCN1. We propose that the coupling mechanism involves the reorganization of the interfaces between the VSD helices, in particular S4, and the pore-forming helices S5 and S6, subtly shifting the balance between hydrophobic and hydrophilic interactions in a 'domino effect' during activation and gating in this region. Remarkably, our simulations reveal state-dependent occupancy of lipid molecules at this emergent coupling interface, suggesting a key role of lipids in hyperpolarization-dependent gating. Our model provides a rationale for previous observations and a possible mechanism for regulation of HCN channels by the lipidic components of the membrane.

**\*For correspondence:**
lucie.delemotte@scilifelab.se

[†]These authors contributed equally to this work

## Editor's evaluation

In this study the authors aim to describe the electromechanical coupling responsible for activation of a Hyperpolarised-activated and Cyclic Nucleotide-gated (HCN) channel. HCN channels are the only mammalian channels to open under hyperpolarisation, being important for their roles in cardiac and neuronal cells. The authors use enhanced-sampling atomistic simulations to enforce sampling between open and closed states of the channel. The simulations suggest state-dependent interactions involving pore and voltage sensor helices, as well as with lipids, leading the authors to propose a domino-like mechanism of activation. These findings will be of considerable interest to the ion channel community.

## Introduction

Hyperpolarized-activated and cyclic nucleotide-gated (HCN) channels, also known as pacemaker channels, are non-selective cation channels located at the plasma membrane of pacemaker cells in the heart and brain. These channels are crucial to control the rhythmic activity in many cell types and further regulate cellular excitability in a wide range of electrically excitable cells (*Brown et al., 1979*; *Santoro et al., 1998*). The ability to drive rhythmically firing action potentials stems from the unusual and characteristic channel activation under membrane hyperpolarization. Given their ubiquity in excitable organs and unique role in electrical signaling, HCN channels have been implicated in a variety of diseases (e.g. heart arrhythmias, epilepsies, and neuropathic pain) and are considered as promising targets for the development of novel drugs (*Benarroch, 2013*; *Biel et al., 2009*; *Bleakley et al., 2021*; *DiFrancesco and DiFrancesco, 2015*; *DiFrancesco et al., 2019*; *Oyrer et al., 2019*; *Tanguay et al., 2019*).

HCN channels are members of the voltage-gated ion channel (VGIC) superfamily, and as such form tetrameric assemblies around a central pore that opens and closes in response to changes in the transmembrane voltage. Each monomer is composed of six transmembrane helices divided into two domains: the voltage-sensing domain (VSD) is a four-helical bundle formed by the first four helices (S1–S4) (*Long et al., 2005*) while the pore domain is formed by tetramerization of the final two helices (S5 and S6) around the symmetry axis (*Lu et al., 2001*).

HCN and other VGICs are allosteric systems where movement of gating charges in the VSD upon changes in electric field governs opening and closing of the pore gate located >10 Å away at the C-terminus of the S6 helix in a process known as electromechanical coupling (*Blunck and Batulan, 2012*; *Long et al., 2005*). Interestingly, the polarity of the coupling in HCN channels is inverted relative to nearly all other VGICs; HCN channels are opened by the downward movement of the gating charges during hyperpolarization, whereas most other channels are closed (*Männikkö et al., 2002*). Inverted electromechanical coupling is central to the role of HCN channels in pacemaking, but the molecular mechanisms underlying inverted coupling have remained elusive.

The first high-resolution structures of HCN1 revealed that, unlike canonical VGICs, HCN channels adopt a non-domain swapped architecture where the VSD packs against the pore domain of the same subunit (*Lee and MacKinnon, 2017*). This unusual fold is not unique to HCN channels but rather is characteristic of the subfamily of channels containing a C-terminal cyclic nucleotide-binding domain (CNBD) connected to the pore via the C-linker (*James and Zagotta, 2018*). Other members of this family include the depolarization-activated KCNH subfamily, such as ether-á-go-go (EAG) and human EAG-related gene (hERG), and voltage-insensitive cyclic nucleotide gated (CNG) channels (*Whicher and MacKinnon, 2016*; *Wang and MacKinnon, 2017*; *Li et al., 2017*).

A flood of new structures in addition to the functional and computational studies they fueled have revealed new atomistic details of voltage sensing and pore gating for numerous members of the CNBD family including HCN channels. HCN1 structures were obtained in resting and activated conformations of the VSD but the pore remained closed in both models (*Lee and MacKinnon, 2017*; *Lee and MacKinnon, 2019*). The model for VSD activation from these structures agrees with predictions from cysteine accessibility (*Bell et al., 2004*), patch-clamp fluorometry (*Dai et al., 2019*), and molecular dynamics (MD) simulations (*Kasimova et al., 2019*). Furthermore, structures of HCN4 have been resolved in both the closed and open state, but in both cases, the VSD is in the resting conformation (*Saponaro et al., 2021*). Together, the HCN1 and HCN4 structures represent three out of the four states in a classical allosteric model of channel gating; the resting-closed, activated-closed, and resting-open conformations. The missing activated-open state is central to understanding the electromechanical coupling mechanism in the HCN family.

Unfortunately, even with a structure of the activated-open state in hand, the weak nature of the VSD-pore coupling in these channels limits our ability to make inferences about the coupling pathway. The experimentally measured coupling energy of only 3–4.5 kcal/mol indicates that activation of all four voltage sensors only increases the stability of the open pore relative to the closed pore by roughly one hydrogen bond (*Ryu and Yellen, 2012*). Given this weak coupling, it is important to not only examine the static structures themselves, but also the structural ensembles and dynamics of the interactions comprising each state.

In this work, we use a stepwise approach to build an understanding of the electromechanical coupling pathway in HCN1. First, we characterize state-specific interactions by analyzing MD

simulations of existing cryoEM structures under equilibrium and activating conditions. We then use a small subset of the interactions likely to stabilize the activated-open channel in enhanced sampling simulations to drive the electromechanical coupling process. Gathering the results, we propose an electromechanical coupling model that relies on a fine-tuned balance between hydrophilic and hydrophobic interactions at the voltage-sensor and pore domain interface. Notably, by characterizing the state-dependent interactions between lipids and the channel domains, we propose an important role for amphiphilic membrane components, rationalizing the effect of changes in membrane composition on the function of this channel.

## Results

### State-dependent interactions between VSD and pore domain in HCN1 models

In the current model for VSD activation in HCN channels, hyperpolarization triggers a downward motion of S4 coupled to a kinking of the helix at S272. This breaks S4 into two sub-helices with the lower segment becoming almost parallel to the membrane (*Figure 1A and C*). This non-canonical voltage sensing mechanism is supported by both experimental and computational studies combined with the cryoEM structure of the activated state stabilized by cysteine cross-linking (*Lee and MacKinnon, 2017*; *Lee and MacKinnon, 2019*; *Vemana et al., 2004*; *Bell et al., 2004*; *Dai et al., 2019*; *Kasimova et al., 2019*). An additional structure was resolved for the functionally locked-open Y289D mutant, in which the gating charges are in a resting state and the pore is closed, but the lower S5 helix swivels away from S6 relative to the position in the wild-type channel in a manner similar to the activated state structure (*Figure 1B*; *Lee and MacKinnon, 2019*). While the mechanism of VSD activation is well established, the mechanism by which this voltage sensor movement triggers pore opening remains poorly understood as the activated-open state of HCN has not been determined.

The movement of the lower half of the S4 helix upon activation drastically alters the interaction networks between the VSD, N-terminal HCN domain, and pore (*Figure 1A–C*). To help understand which state-dependent interactions stabilize each of the conformations observed in cryoEM, we carried out 1-µs-MD simulations of the resting, activated, and Y289D mutant conformations under equilibrium conditions (0 mV).

A comparison between structures of the resting, activated, and Y289D mutant structures (*Figure 1D*), preliminary exploratory MD simulations, and a putative important role of the W281-N300 contact in determining the polarity of HCN1 coupling (*Ramentol et al., 2020*) highlighted W281 in the lower half of S4 as a potentially critical residue in gating, prompting us to further analyze its state-dependent interaction pattern. In the resting state, W281 is facing S1 and nestled in a hydrophobic cavity formed at the interface between S4, S1, and the HCN domain (*Figure 1D*) and remains stably bound at this position throughout the simulation (*Figure 1G*, blue bars). This conformation of W281 is stabilized by extensive hydrophobic interactions with S1 and the HCN domain in addition to an electrostatic contact with S1 for >50% of the simulation time (*Figure 1G*).

In the activated state, the intracellular S1-S4-S5 region becomes much less compact and W281 is pulled out of the hydrophobic cavity, disrupting all interactions with S1 (*Figure 1F*). A new interaction with the pore domain is formed via V296 in S5, but the overall protein contact of W281 is significantly reduced in the activated state. As a result, W281 primarily interacts with lipids (POPC) in the activated state through hydrophobic and electrostatic contacts.

The side-chain orientation and contacts formed by W281 in the Y289D mutant retain aspects of both the resting and activated states (*Figure 1E*). While W281 remains close to S1, the interaction network with S1 is significantly altered relative to the resting state. Furthermore, the strong interaction with the HCN domain enclosing W281 in a hydrophobic cavity in the resting state is disrupted in the mutant. These differences stem from the rotation W281 to an orientation more similar to the activated state of the VSD. Rotation of W281 enables increased contacts with lipids and the pore domain similar to the activated state but interaction with S5 occurs via N300 in addition to the V296 contact observed in the activated state.

As a result of the mixed set of W281 contacts shared with the resting and activated states, the structure of the Y289D mutant can be described as an 'intermediate state' conformation for VSD activation. The rotation of W281 resembles the activated state, but the interactions with S1 retain the

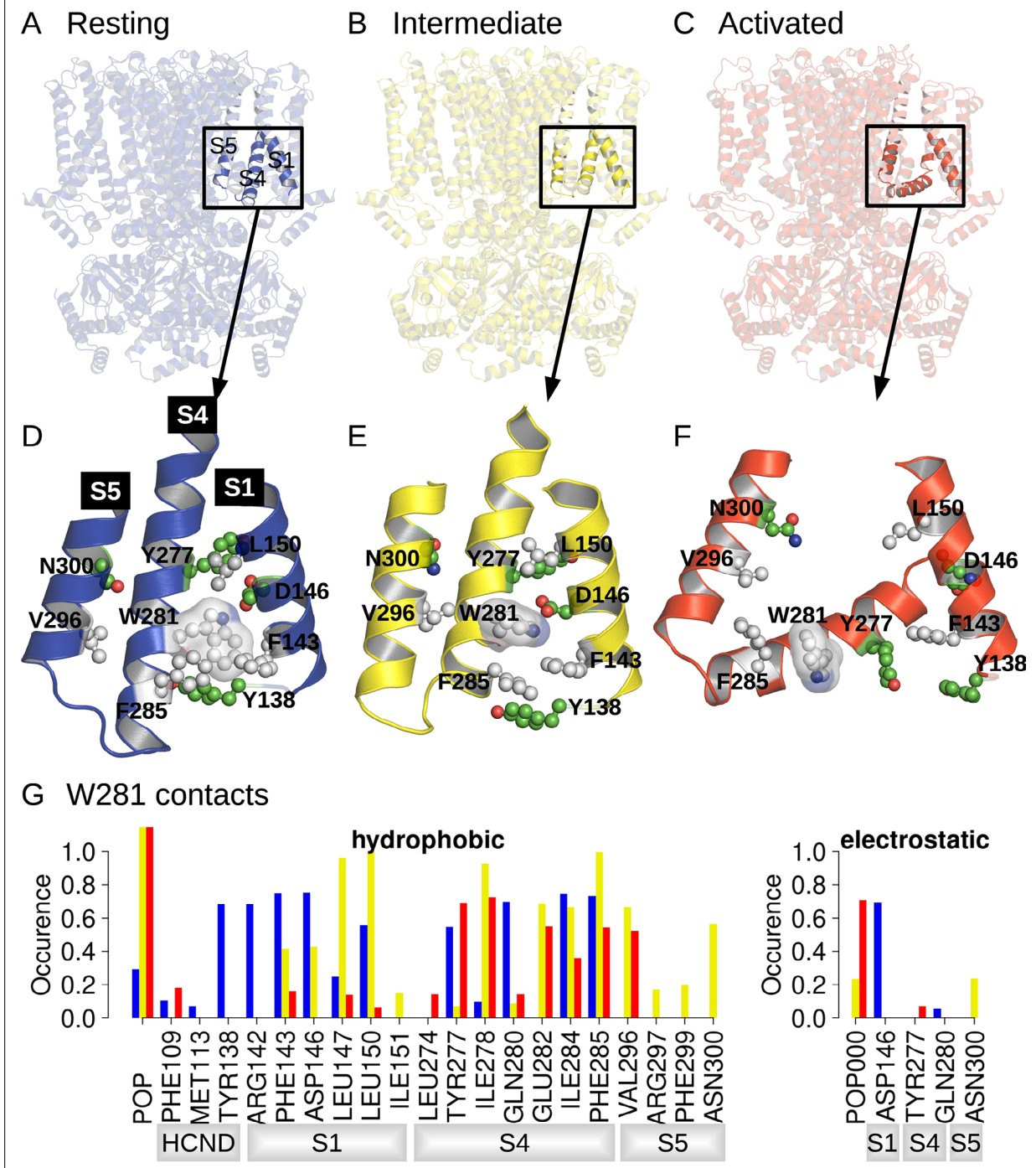

**Figure 1.** Position and contact network of W281. The three cryoEM structures of the resting (**A;** PDB ID 5U6P), intermediate (**B;** PDB ID 6UQG), and activated (**C;** PDB ID 6UQF) states of hHCN1, shown in blue, yellow, and red, respectively, were investigated by equilibrium molecular dynamics (MD) simulations. The position of W281 is highlighted in ball/stick, and transparent surface and surrounding residues are shown in ball and stick in the resting (**D**), intermediate (**E**), and activated (**F**) conformations. (**G**) Contact occurrence between W281 and nearby residues and lipid molecules along 1-μs-long MD simulations.

The online version of this article includes the following figure supplement(s) for figure 1:

**Figure supplement 1.** S4-S5 and S4-S1 hydrophobic interfaces in the resting, intermediate, and activated states.

compact nature of the S1-S4-S5 region similar to the resting state. This configuration is quite interesting as the Y289D mutant retains many characteristics of the VSD-pore interactions of the activated state despite the fact that the gating charges are in their resting position and S4 does not adopt the characteristic bend associated with voltage sensor activation.

## Dynamics of VSD-pore interactions upon voltage sensor activation

The state-dependent VSD-pore interactions correlated with rotation of the C-terminus of S4 at W281 outlined above hint at a possible functional role in electromechanical coupling. To gain insight into the evolution of these state-dependent interactions over the course of channel activation, we simulated the resting state under a hyperpolarizing transmembrane potential. We used three new strategies in an attempt to further accelerate VSD activation compared to our previous study in hopes of driving the channel to the activated-open state. First, we started from the cAMP-bound configuration of the CNBD, which has previously been shown to allosterically promote both pore opening and VSD activation (*Wu et al., 2011*; *Kusch et al., 2010*). Next, we increased the magnitude of the membrane potential to –1 V (compared to –550 mV previously used). This should reduce the time constant for activation from 23 µs to 3 µs based on our previous kinetic analysis (*Kasimova et al., 2019*). Finally, we simulated an additional system lacking the HCN-domain (called hereafter HCND-less system), which has previously been suggested to inhibit VSD activation (*Saponaro et al., 2021*). This is consistent with our results above where the HCN domain acts as a lid enclosing W281 in a hydrophobic cavity in the resting state.

In the intact system, full activation occurs in only one VSD as marked by movement of R267 past the charge transfer center (*Figure 2A and B* and *Figure 2—figure supplement 1A*). Partial activation as seen by movement of R270 past the charge transfer center occurs in one subunit while the remaining two VSDs show little to no displacement. In the subunits showing full or partial activation, S4 adopts the characteristic bend associated with channel activation observed in our previous simulations and the cryoEM structure of the cadmium-crosslinked activated state (*Figure 2—figure supplement 1A*). This suggests that the activation mechanism is preserved despite the high magnitude of transmembrane potential used. The incomplete activation of the voltage sensors in this system is not surprising given the length of the simulation (1 µs) compared to the predicted time constant for activation (3 µs).

For the HCND-less system, one VSD undergoes full activation while the remaining three undergo partial activation over the course of a 750 ns simulation (*Figure 2D and E* and *Figure 2—figure supplement 1B*). This is consistent with an accelerated voltage sensor activation compared to the intact system expected by relieving inhibition of activation provided by the HCN domain. However, the wider distribution of bending angles in S4 for the system lacking the HCN domain indicates that the HCN domain may also contribute to stabilizing the activated state of S4 (*Figure 2C and F*).

During VSD activation, downward movement of the gating charges and bending of S4 is correlated with rotation of the C-terminus of S4, pointing W281 toward the lipid interface (*Figure 2—figure supplement 1* and *Figure 2C and F*). In subunits undergoing complete activation, a more complete rotation of the C-terminus of S4 results in the formation of a hydrogen bond between W281 and N300 that was observed in the Y289D mutant above (*Figure 2A and D* and *Figure 2—figure supplement 2A and D*), consistent with equilibrium simulations of the activated state of HCN1 (*Figure 1—figure supplement 1B*). Interestingly, swapping these two residues inverts the gating polarity in the related spHCN channel (W355N/N370W, spHCN numbering). The proposed basis of this inverted coupling was a similar rotation of the lower S4 helix as we observed, but driven by the introduction of a hydrophobic mismatch of the mutations rather than activation of the voltage sensor (*Ramentol et al., 2020*).

In our simulations of VSD activation, rotation of W281 is accompanied by an increase in the hydrophobic contact area at the lower S4/S5 interface with little change to the electrostatic interactions (*Figure 2G* and *Figure 2—figure supplement 2B and E*), reminiscent of the behavior observed in equilibrium simulations of the resting and activated states (*Figure 1—figure supplement 1A*). On the other hand, analysis of the lower S4-S1 interface showed a decrease in the hydrophobic contacts between S4 and S1 in most (seven out of eight) subunits (*Figure 2—figure supplement 2C and F*). This suggests that separation of the S1/S4 interface is one of the first steps toward the release of W281 from the S1/S4/HCN domain hydrophobic pocket and enabling rotation of the C-terminus of S4.

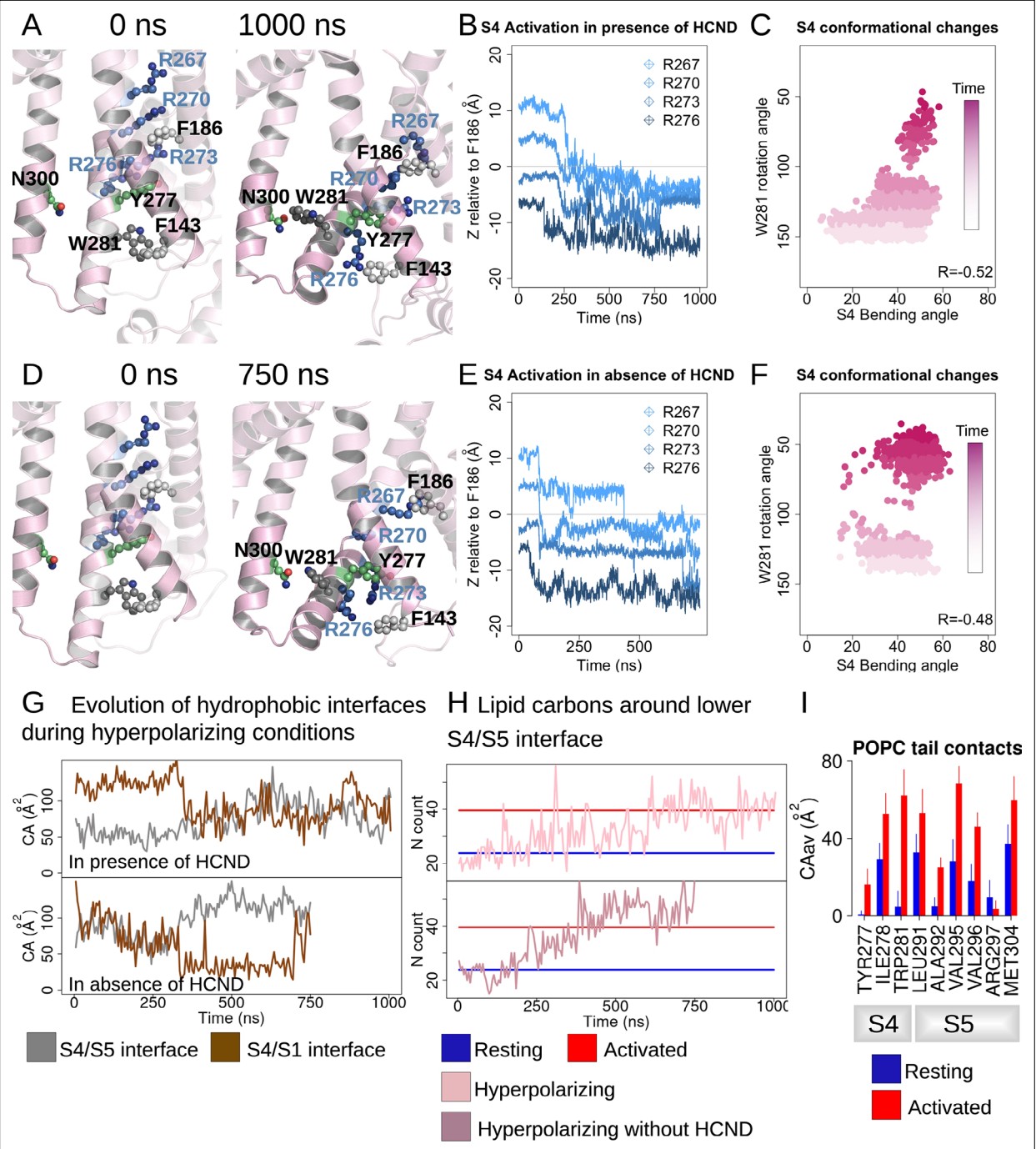

**Figure 2.** HCN1 activation induces an increase in the lower S4/S5-lipids hydrophobic interfaces. (**A, D**) First and last snapshots of voltage-sensor domains (VSDs) extracted from activating simulations of intact (panel **A**, subunit D) or HCND-less system (panel **D**, subunit A). The gating charges and residues W281, Y277 (S4), N300 (S5), and F143 (S1) are highlighted using ball and stick representation. (**B, E**) Downward displacement of the gating charges with respect to the charge transfer center F186 during the activation of S4 in subunit D (panel **B**, intact HCN1 model) and subunit A (panel **E**, HCND-less HCN1 model) during activating simulations. (**C, F**) Correlation between the S4 bending angle and the rotation of W281 in subunit D (panel **C**, intact HCN1 model) and subunit A (panel **F**, HCND-less HCN1 model) during activating simulations. The time evolution is represented by a color gradient, from white/light pink to dark pink. (**G**) Evolution of the S4/S5 (brown) and S4/S1 (gray) hydrophobic interfaces (contact areas) along activating simulations of HCN1 in the presence (subunit D, top box) or absence (subunit A, bottom box) of the HCN domain. (**H**) Evolution of the number of lipid carbon atoms, N, in contact with hydrophobic residues from the lower S4/S5 interface along activating simulations of HCN1 in the presence and absence of HCN domain are shown for subunit D in light pink (top box) and for subunit A in dark pink (bottom box), respectively. N count values for the resting and activated states were averaged for the four subunits from the equilibrium simulations and are displayed as blue and red horizontal lines,

*Figure 2 continued on next page*

*Figure 2 continued*

respectively. (**I**) Contact area between lipid tails and residues on the lower S4/S5 interface, averaged over the four subunits and over the simulation time, in the resting (blue) and activated (red) states. Error bars are given as standard deviation calculated over the four subunits and over the simulation time.

The online version of this article includes the following figure supplement(s) for figure 2:

**Figure supplement 1.** S4 displacement in simulations of the intact (**A**, 1 µs) or in the HCND-less system (**B**, 750 ns).

**Figure supplement 2.** Evolution of the S4/S5 and S4/S1 interfaces in activating molecular dynamics (MD) simulations of intact (**A–C**) and in HCND-less HCN1 models (**D–F**).

**Figure supplement 3.** Hydrophobic and electrostatic interactions in the lower S4/S5 region are state-dependent.

The combined bending and rotation of the lower S4 accompanies the displacement of S1 away from S4 and exposes the VSD-pore interface to the lipid bilayer. As a result, an increased number of contacts between lipid tails and hydrophobic residues on S4 and S5 accompanies VSD activation (*Figure 2I* and *Figure 2—figure supplement 3*). The average number of POPC carbon atoms in contact with the lower S4/S5 interface is ~24 ± 7 in the resting state versus ~40 ± 7 in the activated state equilibrium simulation. This increase in lipidic contact upon activation is preserved whether the HCN domain is present or absent. Thus, lipid tails preferentially contact the VSD/pore interface in the activated state of the voltage sensor, indicating that lipid tails may play a role in electromechanical coupling in HCN1. This may help reconcile the fact that the cryoEM structure of HCN1 with an activated voltage sensor as well as the Y289D mutant in detergent remained closed despite functional evidence suggesting that these channels should remain open at 0 mV (*Lee and MacKinnon, 2019*).

Despite the combination of three strategies to accelerate activation, the pore gate remained shut throughout these simulations. This is unsurprising given the slow intrinsic gating of the pore and the weak electromechanical coupling in the HCN family. Even in the presence of four activated voltage sensors, the closed to open pore transition for HCN1 has a time constant of ~100 ms based on allosteric models of channel gating (*Altomare et al., 2001*). Consequently, the pore opening process is far outside the current range of unbiased MD simulations for HCN1. We thus set out to further analyze state-dependent interactions in an attempt to pinpoint those that may help favor pore opening, to then be able to use them in enhanced sampling simulations.

## Displacement of S5 during VSD activation alters interaction between S4-S5 linker and C-linker

As a result of the bending and rotation of S4 during VSD activation, the N-terminal end of S5 is displaced relative to the resting state conformation in a movement that was previously observed in MD simulations and cryoEM structures. Interestingly, the N-terminus of S5 is similarly displaced in the Y289D mutant cryoEM structure despite lack of VSD activation in this channel (*Figure 3A*). These tilted conformations of the S5 helix are also observed in open-channel structures of related CNBD family channels like hERG and TAX-4 (*Wang and MacKinnon, 2017*; *Zheng et al., 2020*). Furthermore, this region of S5 directly contacts the S6 helix near the activation gate, thus the tilting movement of S5 following VSD activation was proposed as a critical step in the electromechanical coupling of HCN1 (*Lee and MacKinnon, 2019*; *Kasimova et al., 2019*) and experimentally shown in HCN4 (*Saponaro et al., 2021*).

In our simulations of the resting, activated, and intermediate (Y289D) structures above, we noted that tilting of S5 also alters the interaction network between the S4-S5 linker and the C-linker of the channel (*Figure 3B*). The C-linker directly connects the CNBD to the S6 activation gate and has previously been identified to play a role in electromechanical coupling based on functional studies (*Dai et al., 2021*); therefore, we suspected these activation-induced interactions may help drive channel opening.

The interaction between D290 of the S4-S5 linker and K412 of the C-linker is of particular interest given the clear state-specificity to both the active and intermediate states. The absence of this interaction in the resting state stems in part from the large distance between the positions of the backbones of these residues (*Figure 3B*). The tilting of S5 caused by activation or the Y289D mutation brings the D290-K412 positions in close proximity and enables stable salt bridge formation.

Interestingly, the D290-K412 distance in the original model of the activated VSD-closed pore conformation is close to the distance observed in the resting state structure. However, this distance

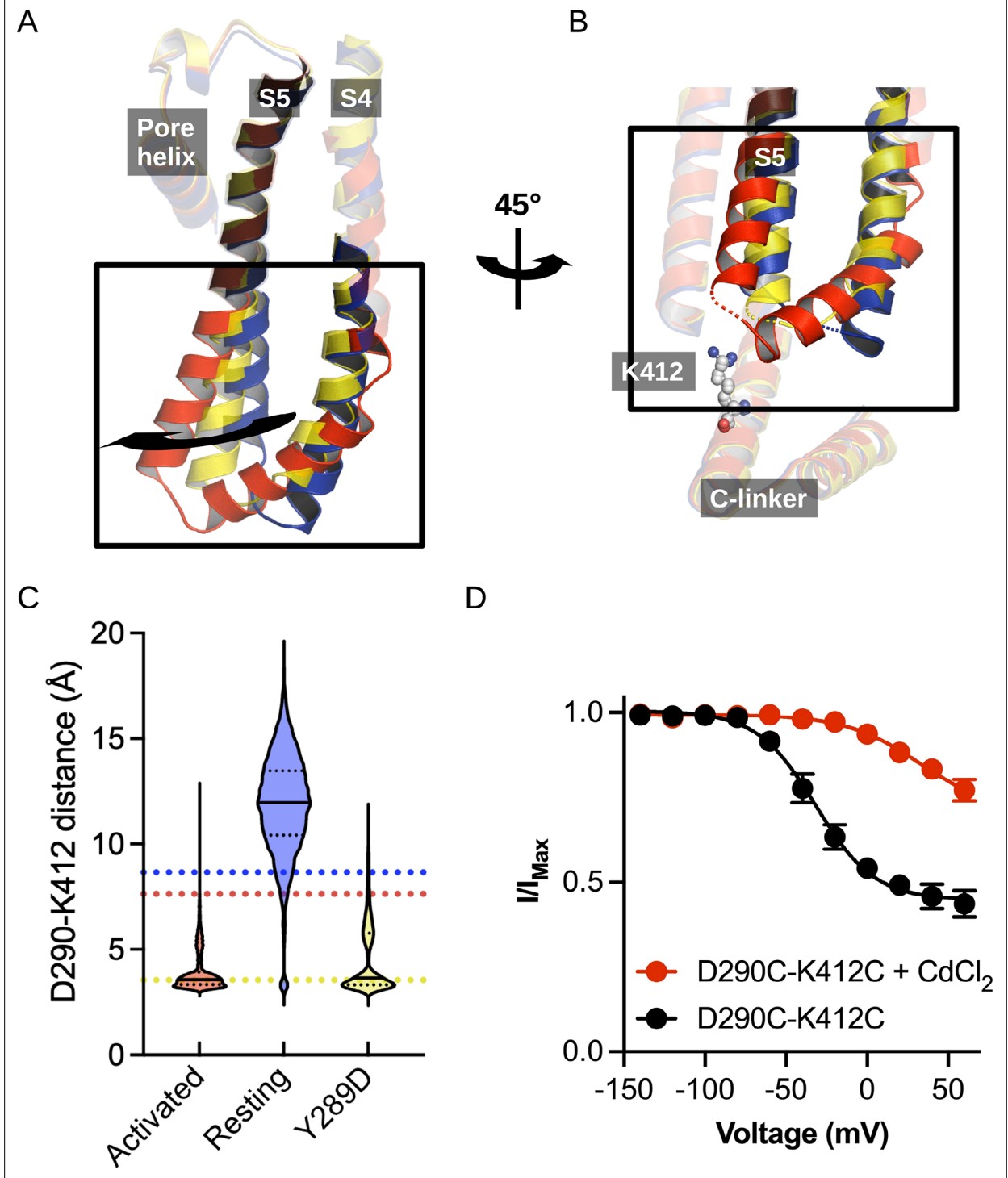

**Figure 3.** Structural comparison of the S4/S5 linker and C-linker regions. (**A**) Structural comparison of the S4-S6 region of the resting (5U6P; blue), intermediate (6UQG; yellow), and activated (6UQF; red) states of hHCN1. Alignment was performed on the entire structure. (**B**) Structural comparison of the S4/S5 linker and C-linker regions of the resting, intermediate, and activated states of hHCN1. Residue K412 of the C-linker is shown as grey spheres. (**C**) D290-K412 distance distribution in simulations of the resting, intermediate, and activated states. Dashed lines represent distances in the starting cryoEM structures of the same color. (**D**) Peak tail current versus voltage plot for whole-cell recordings from HEK cells expressing HCN1 D290C/K412C with (red) or without (black) 100 uM CdCl$_2$ in the patch pipette. Error bars represent SEM for n = 5 independent cells for each condition.

The online version of this article includes the following figure supplement(s) for figure 3:

**Figure supplement 1.** Functional recordings of the D290C-K412C double mutant.

decreases rapidly in the first few picoseconds of simulations of the activated VSD-closed pore structure and the resulting interaction remains stable for the majority of the trajectory (*Figure 3C*). We thus hypothesize this interaction is involved in coupling VSD movement to gate opening through the C-linker.

Given that this interaction was not observed in the cryoEM structure of the activated VSD, we sought to experimentally validate the D290-K412 interaction in the activated-open state of the channel. Cysteine-crossbridging has been a successful approach to probe state-dependent interactions in HCN channels (*Rothberg et al., 2003*; *Lee and MacKinnon, 2019*). In this technique, pairs of residues are mutated to cysteine and then exposed to cadmium, which will form high-affinity bonds between the thiolate groups provided the distance between residues is in the range of 6–8 Å. If the suspected interaction is state-specific, it will 'lock' the channel in the corresponding functional state.

The HEK cells expressing D290C-K412C double mutant in the absence of cadmium conduct a relatively high basal current and activate at depolarized potentials compared to the HCN1-EM parent constructs (*Figure 3—figure supplement 1A*). Much of this basal current can be blocked by application of external cesium chloride, indicating that the D290C-K412C mutant does not fully close even in the absence of applied cadmium (*Figure 3—figure supplement 1B and C*). Disrupting the D290-K412 interaction is expected to destabilize the open state based on our simulations above, in contrast to the observed rightward shift in the activation curve and increase in basal activity we observe in the D290C-K412C mutation. The enhanced apparent stability of the open state in the D290C-K412C mutant may be due to spontaneous disulfide bond formation during the patch experiment or from disruption of closed-state-specific interactions of these residues.

In HEK cells patched with 100 uM cadmium chloride in the pipette, the basal conductance increases substantially to $0.77 \pm 0.7$ compared to $0.44 \pm 0.4$ in the absence of cadmium (*Figure 3D*). Furthermore, the voltage dependence of activation shifts by 60 mV to more positive potentials (*Figure 3D*). This is not observed in the wild-type channel in the presence of cadmium, which only displays a minor rightward shift in the activation curve (*Figure 3—figure supplement 1A*). The combined increase in the basal current amplitude and rightward shift in the voltage-dependence of activation in the double cysteine mutant in the presence of cadmium strongly indicates that crossbridging these positions preferentially stabilizes the open state, in agreement with our hypothesis that the D290-K412 interaction promotes channel opening.

## Driving electromechanical coupling in enhanced sampling simulations

In all of our simulations described above, the S6 activation gate remained tightly closed and dehydrated for the entirety of the simulation. To gain insights into gating transitions of the HCN1 pore, we turned to enhanced sampling simulations in which pore opening was encouraged by the application of biasing forces. In previous sections, we had identified several interactions that are likely involved in stabilizing the activated voltage sensor and/or open pore. The W281-N300 pair at the S4/S5 interface stabilizes the rotation of the C-terminus of S4 on activation. Interaction between D290-K412 on the S4-S5 linker and C-linker promotes the bending of S5 away from the resting conformation, loosening the packing against the closed state of S6. In addition, a structural comparison of pairs of closed and open structures of CNBD channels revealed that distances between V390 consistently increased upon pore opening in TAX-4; this distance increases from 10.6 to 17.2 Å (PDB IDs 6WEJ-6WEK), in CNGA1 from 9.6 to 16.4 Å (PDB IDs 7LFT-7LFW), in CNGA1/B1 from 9.8 to 15.1 Å (PDB IDs 7RH9-7RHH), in EAG/hERG from 10.7 to 17.3 Å (PDB IDs 5KL7L-5VA1), and in HCN4 from 10.5 to 15.0 Å (PDB IDs 7NP4-7NP3). Using these state-specific distances as collective variables (CVs) in enhanced sampling MD simulations (*Figure 4A*), we hypothesized that we could drive the electromechanical coupling process in a computationally accessible timescale.

Of the numerous available enhanced sampling techniques, we opted for adiabatic bias MD (ABMD) simulations given the gentle nature of the biasing potential applied (*Hénin et al., 2022*). ABMD simulations allow thermal fluctuations to carry the system forward over energy barriers while only applying biasing potentials dampening movement in the backward direction. We introduced these ratchet-like biases to encourage the formation of interactions between N300-W281 and D290-K412 and discourage interactions between V390-V390 cross- and neighboring pairs, in 1 μs ABMD simulations starting from the HCN1 resting/closed state (two replicates) and from the HCN1 activated/closed state (two replicates).

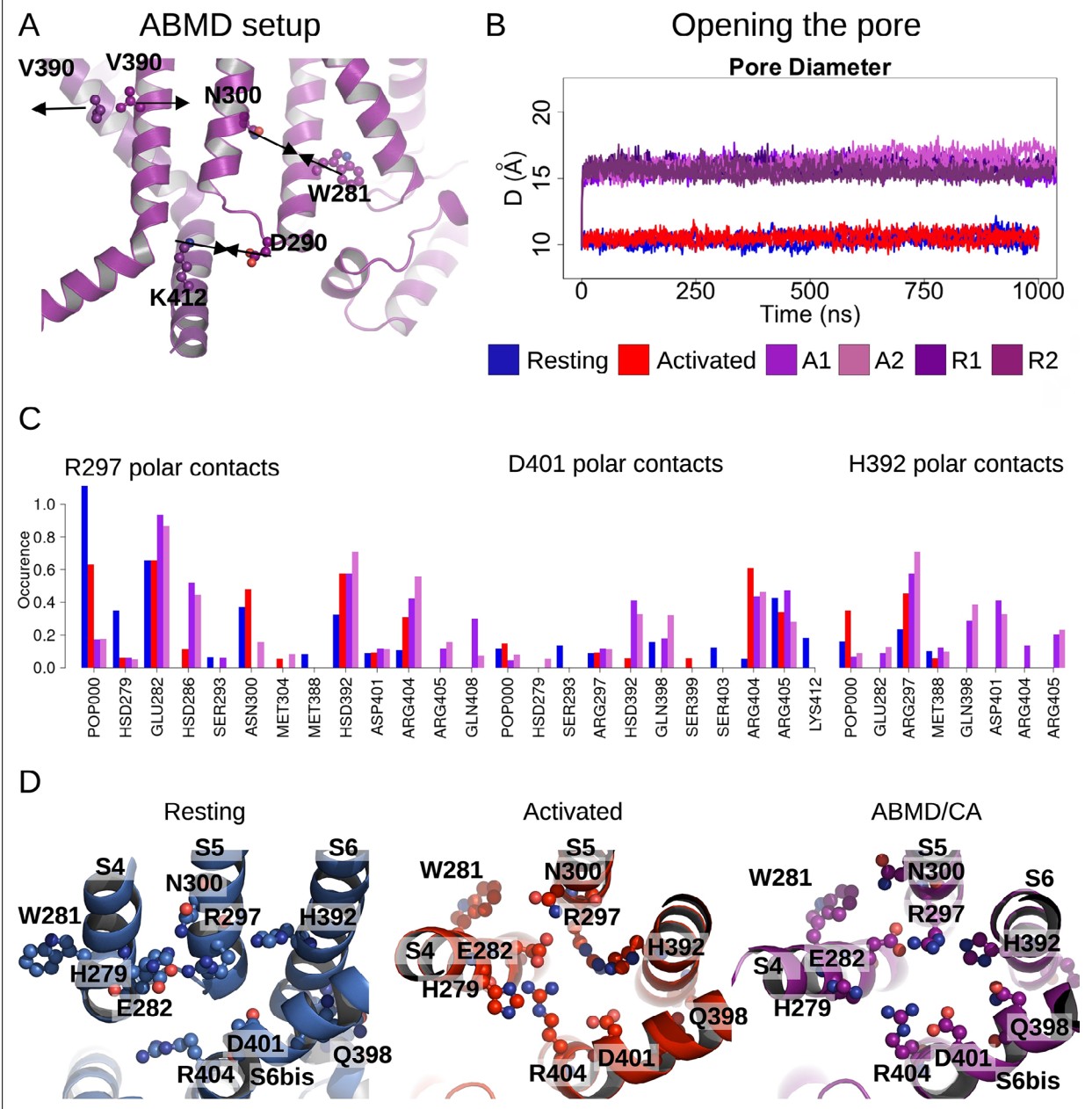

**Figure 4.** Pore opening of HCN1. (**A**) Residues and distances used as collective variables in adiabatic bias molecular dynamics (ABMD) simulations are highlighted by ball and stick and black double arrows, respectively. (**B**) Inter-subunit distances between opposite V390 residues are represented as a function of time in the resting (blue), activated (red), and ABMD (crimson) simulations starting from the closed activated (A1/A2 in light purple and light orchid) and resting (R1/R2 in dark purple and dark orchid) states. (**C**) Contact occurrences of R297, D401, and H392 in HCN1 are shown for the resting (blue), activated (red), and ABMD (crimson) simulations starting from the closed activated (A1/A2 in light purple and light orchid). (**D**) Lower S4/S5, S5/S6, S6/S6(bis) S4/CL(bis) interfaces/interactions. Important/interesting residues from these interfaces are highlighted in ball and stick.

The online version of this article includes the following figure supplement(s) for figure 4:

**Figure supplement 1.** Open pore models from adiabatic bias molecular dynamics (ABMD) simulations compared to HCN4.

**Figure supplement 2.** Potassium permeation along the HCN1 pore.

**Figure supplement 3.** Key state-dependent interactions.

**Figure supplement 4.** Opening implies slightly reinforced hydrophobic S6-S5 interfaces and substantially weakened hydrophobic S6-S6 interfaces.

During these simulations, we observed a splaying apart of the S6 helices, with a hinge forming around G382, and a rotation and slight deformation of the helix bringing V390 from a position facing the pore lumen to a position still lining the inner vestibule in the open state (*Figure 4—figure supplement 1*). This event is coupled to an increase in the pore radius at the V390 position (*Figure 4B*) and an increased hydration of the pore intracellular gate, enabling the conduction of several K$^+$ ions (1–3 ion permeation events in each simulation, *Figure 4—figure supplement 2*). The rearrangement of S6 also induced the formation of other interactions between adjacent S6 helices: in particular, a contact between H392 and adjacent Q398 formed during the ABMD simulation initiated in the activated/closed state, even though the formation of these contacts was not explicitly encouraged in the biasing simulations (*Figure 4C and D* and *Figure 4—figure supplement 3*). The contact between positions N658-Q664 (equivalent to H392-Q398) is also present in the open state structure of hERG (*Wang and MacKinnon, 2017*). Our data also suggest that other contacts could be specifically formed in the open channel, like between E282 from S4 and R404 from the adjacent C-linker. Thus, we suggest that the H392-Q398 contact (and possibly of the other aforementioned contacts with E282 and R404) should be formed to induce pore widening and stabilize the open conformation in HCN1.

## Energetic basis for electromechanical coupling

Calculations of the intra-subunit contact areas between S6 (from residue 386–401) and S5 (lower segment) showed a slight reinforcement of the hydrophobic contact area in the open model and ABMD simulations of HCN1, indicating that closure/opening may be related to changes of interfaces between pore-forming helices (*Figure 4—figure supplement 4A*). By contrast, the adjacent S6-S6 hydrophobic interfaces are substantially weakened, suggesting that the enlargement of the pore radius requires an increased distance between S6 helices (*Figure 4—figure supplement 4B*). In addition, the formation of specific interactions between polar/charged residues at the S4/S5/S6 interface accompany pore opening, most notably between R297 (S5) and E282 (S4) and between R297 (S5) and H392 (S6), which is coupled to a loss of contact of interactions between R297 and lipid headgroups (*Figure 4C and D*). On top of this, a specific interaction between E282 (S4) and R404 (C-linker) appears important to stabilize the open state (*Figure 4—figure supplement 3*).

The analysis of the interaction between lipid headgroups of the lower leaflet POPC lipids and the innermost S4/S5/S6 region also revealed a state-dependent pattern (*Figure 5A*). Indeed, lipids tended to bind in two separate sites (sites A and B, *Figure 5B and C*), located on opposite sides of the S4/S5/S6 interface. In both of these sites, contacts with polar or charged residues were increased in the activated state relative to the resting one and involved a larger number of lipids. Site B specifically engaged VSD residues, including S4 gating charges. Site A, located on the opposite side of the VSD/pore interface, on the other hand, engaged in addition residues on S5 (R297) and on S6 (H392). During pore opening in the ABMD simulations initiated in the activated state, the interactions between the innermost S4/S5/S6 region and lipid headgroups remained largely identical to those pinpointed in the activated state. The only large reduction in interaction concerned R297, which interacted strongly with POPC headgroups in the resting state, substantially less in the activated/closed state and almost lost all contacts in the activated/open state. Interestingly, H392, pinpointed as crucial for pore gating above, is located in the middle of the sequence corresponding to the inner constriction on S6 and is highly conserved in the HCN family. Its position at the lipid interface allows it to be flexible. Thus, this residue is capable of making contacts both with R297 on S5, and with a POPC headgroup from the lower bilayer leaflet (*Figure 5*), sometimes engaging in a ternary interaction with both binding partners. In fact, both the H392-R297 contact and the contact between H392 and lipid headgroups are reinforced in the activated state, while the contact between H392 and lipids is again broken in the open state, confirming that these contacts are likely involved in activation and opening (*Figures 4E and 5*).

We hypothesize that these lipids play a particular role in coupling S5 and S6 in a way as to enable pore gating. Overall, these observations imply that activation and following opening of HCN1 are also driven by a combination of forces: on the one hand, hydrophobic forces reinforce the intra-subunit S4/S5/S6 interface, while on the other hand, the positioning of lipid headgroups in contact with hydrophilic residues located at this interface also guides the positioning of lipid tails that contribute to the overall hydrophobic region.

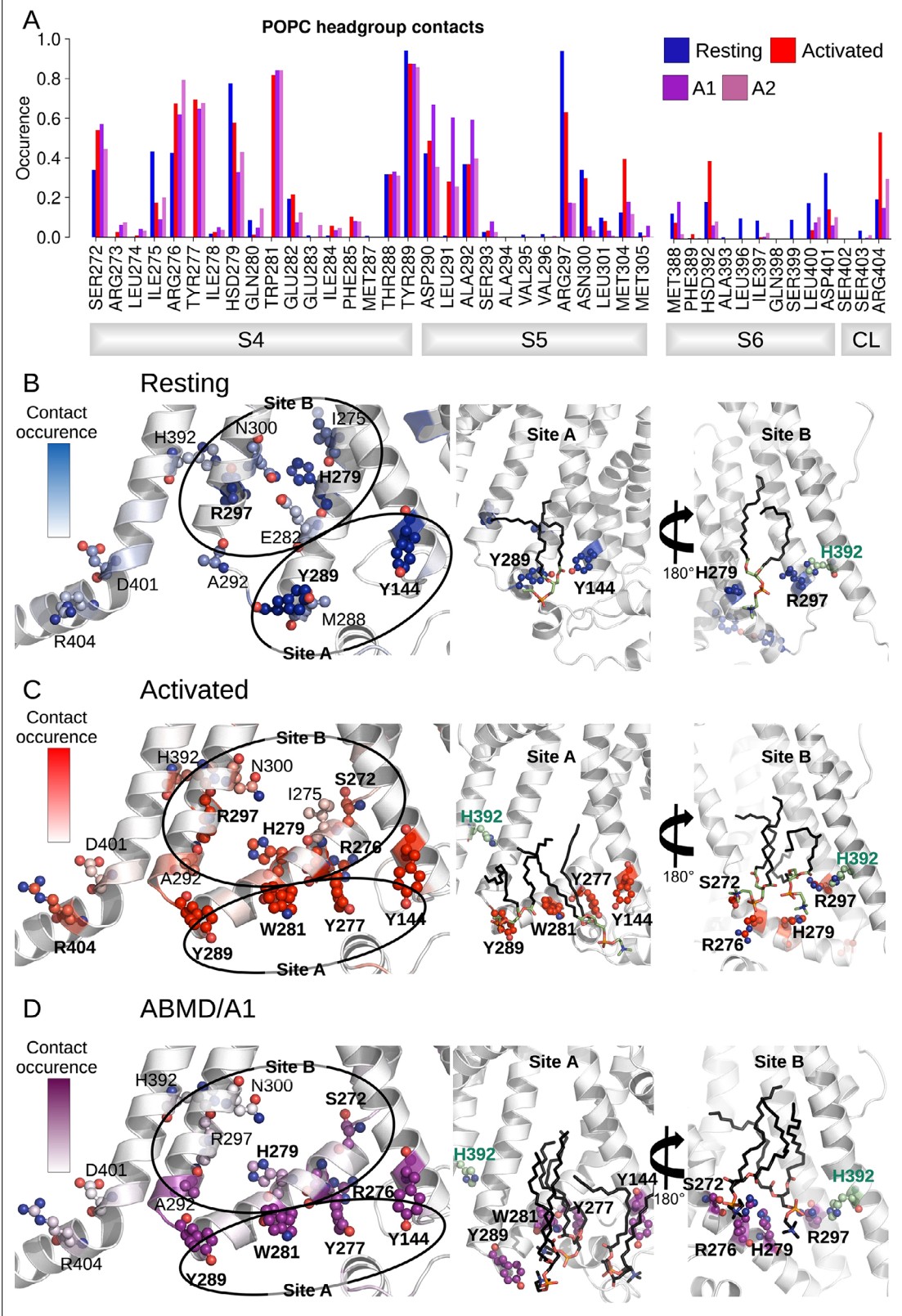

**Figure 5.** Interactions between the lower S4/S5/S6 region of HCN1 and the lipid headgroups are state-dependent. (**A**) Contact area between lower S4/S5/S6 protein residues and lipid headgroups averaged over the four subunits and over the simulation time, in the resting (blue), activated (red), and adiabatic bias molecular dynamics (ABMD) simulations (A1/A2 in purple and orchid) starting from the activated state. (**B**) Residues from lower S1/S4/S5 in contacts with lipid headgroups for more than 50% of the simulation time in the resting HCN1 are highlighted in blue. Snapshot of POPC lipids (sticks)

*Figure 5 continued on next page*

*Figure 5 continued*

bound to lipid headgroup binding sites found in HCN1 resting state (**C**) Residues from lower S1/S4/S5 in contacts with lipid headgroups for more than 50% of the simulation time in the activated HCN1 are highlighted in red. Snapshot of POPC lipids (sticks) bound to lipid headgroup binding sites found in HCN1 activated state. (**D**) Residues from lower S1/S4/S5 in contacts with lipid headgroups for more than 50% of the simulation time in the ABMD simulation A1 are highlighted in purple. Snapshot of POPC lipids (sticks) bound to lipid headgroup binding sites found in ABMD/A1 simulation.

## Discussion

The slow intrinsic gating and weak coupling between the VSD and pore presents a substantial challenge for characterizing the electromechanical coupling pathway in HCN channels with computational methods. While voltage sensor activation can be dramatically accelerated with an electric field, pore opening is only indirectly influenced through allosteric interaction via the electromechanical coupling pathway. Due to the weak coupling energy in HCN channels, the ability of the voltage sensors to expedite gating is much more limited than in strongly coupled VGICs such as Shaker. We have circumvented these limitations using state-specific contacts derived from equilibrium simulations on the existing models of HCN1 to drive the electromechanical coupling process using enhanced sampling. Integrating the results from these simulations enables us to propose a model for electromechanical coupling in HCN channels that builds upon and further refines current models of channel gating (*Figure 6*).

From the resting state of HCN1, hyperpolarization drives downward movement of the gating charges and helix-breakage at S272 and initiates a cascade of rearrangements of the hydrophobic

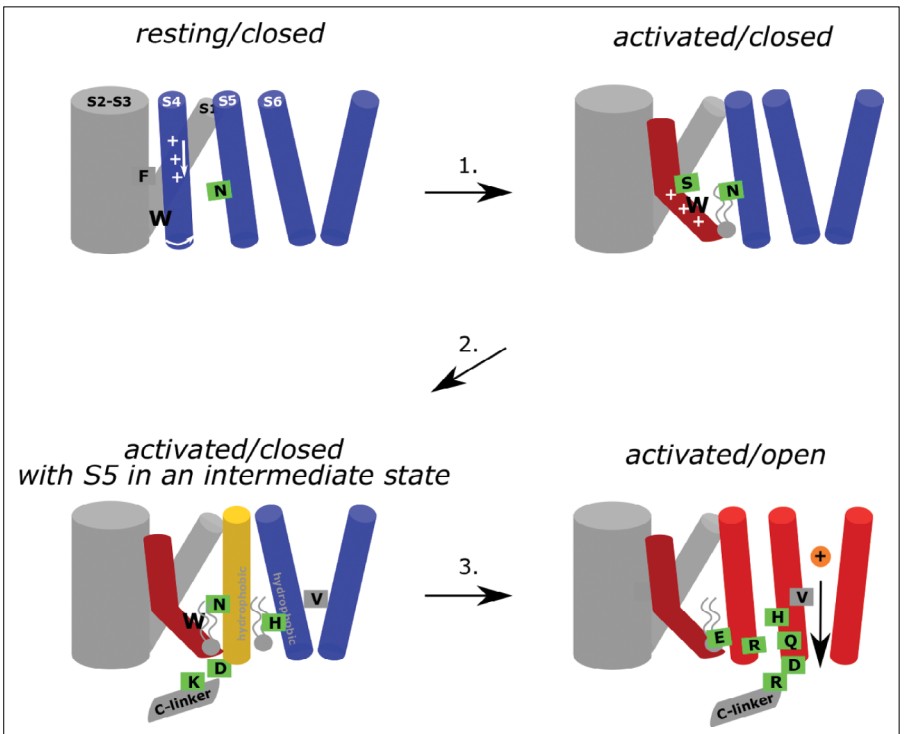

**Figure 6.** Hydrophobic domino-effect model of HCN1 gating upon activation. Hyperpolarization triggers the downward and breaking movement of S4 at S272 and initiates a cascade of rearrangements of the hydrophobic interfaces between and within the voltage-sensor domains (VSD) and pore. First, W281 is pulled out of a S1/HCND hydrophobic cavity leading the lower S4 helix to rotate and form of new contacts with N300 on S5 (step 1). This is accompanied by an increase in the hydrophobic interface between the lower S4, S5, and the surrounding lipid tails. These interactions and the formation of the D290-K412 contact between the S4-S5 linker and the C-linker result in a slight bending of S5 away from S6 as hydrophobic contacts between S5 and S6 form. Lipid contacts at this interface are also increased (step 2). This results in hinging of S6 and a rotation of V390 away from the pore axis, opening of a conductive hydrophilic pathway along the pore axis. The open conformation is stabilized by interactions between R297 (S5) and E282 (S4), between R297 (S5) and H392 (S6), and between H392 and Q398, D401, and R404 from the adjacent subunit (step 3).

interfaces between and within the VSD and pore. This hydrophobic 'domino effect' starts when W281 is pulled out of a hydrophobic cavity formed by S1 and the HCN domain, enabling rotation of the lower S4 helix, driving formation of new VSD-pore contacts between W281 (S4) and V296 and N300 (S5) (*Figure 6*; step 1). This two-step-activation mechanism induces an increase in the hydrophobic interface between the lower S4, S5, and the surrounding lipid tails, thus reinforcing coupling between the VSD and pore as S4 activates. The increased S4/S5 coupling as S4 is displaced from the resting state and the formation of the D290-K412 contact between the S4-S5 linker and the C-linker results in a slight bending of S5 away from S6 (*Figure 6*; step 2) and enables S6 to rotate radially away from the pore axis by hinging at the level of G382 to maintain and even strengthen hydrophobic S5/S6 contacts (*Figure 6*; step 3). Rotation of S6 results in an expansion of the pore via the reorientation of V390 away from the pore axis and into a pore-lining position permitting conduction through the open pore. The combined VSD activation and pore opening also allows a reorganization of the lipid molecules in this region, possibly hinting at the fact that this could represent a crucial lipid modulation binding region.

A two-step mechanism for VSD activation was recently identified in spHCN using VCF and gating currents to track voltage sensor movement on the E356A (*Wu et al., 2021*). This mutant uncouples the primary gating charge movement from pore opening and reveals a secondary movement of the voltage sensor that correlates with channel opening. This secondary movement may correspond to either the bending or rotation of the lower S4 helix. However, our simulation of the structure of the Y289D uncoupling mutant would suggest that rotation of S4 alone can drive pore opening as the W281-N300 contact is formed but no bend is observed in S4, consistent with the model proposed by the Larsson lab (*Ramentol et al., 2020*). Furthermore, this model for an altered S4/S5 interface upon rotation of the lower S4 also explains how voltage sensor movement can still be coupled to pore gating in channels lacking a covalent linkage between the S4 and S5 (*Flynn and Zagotta, 2018*).

Based on their work with such split channels and extensive mutagenesis, Flynn and Zagotta suggested that the lower S4-S5 interface plays a critical role in stabilizing the closed state of the pore (*Flynn and Zagotta, 2018*). Activation of the VSD relieves this inhibitory effect and enables the more favorable open state of S6. Our simulations reveal that pore opening involves a tradeoff between a strengthening of the S5/S6 interface at the expense of a weakening of the S6-S6 interface. This is consistent with previous mutagenesis

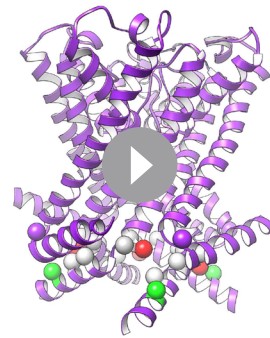

**Video 1.** Structural position of Y289 in ABMD1 open model relative to previously probed positions in the C-linker. ABMD1 open state model highlighting positions of cysteine mutations from *Kwan et al., 2012* highlighted with their beta carbon as spheres. Y289 is colored in the same color as the model while the positions on the C-linker are colored according to their functional effect with applied cadmium. Green indicates lock-open effect, red indicates lock-closed effect, and gray indicates no effect. Only the pore domain and A' helix of the C-linker are shown for clarity. https://elifesciences.org/articles/80303/figures#video1

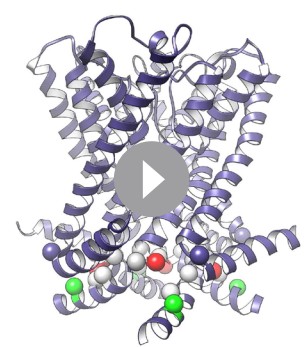

**Video 2.** Structural position of Y289 in ABMD2 open model relative to previously probed positions in the C-linker. ABMD2 open state model highlighting positions of cysteine mutations from *Kwan et al., 2012* highlighted with their beta carbon as spheres. Y289 is colored in the same color as the model while the positions on the C-linker are colored according to their functional effect with applied cadmium. Green indicates lock-open effect, red indicates lock-closed effect, and gray indicates no effect. Only the pore domain and A' helix of the C-linker are shown for clarity. https://elifesciences.org/articles/80303/figures#video2

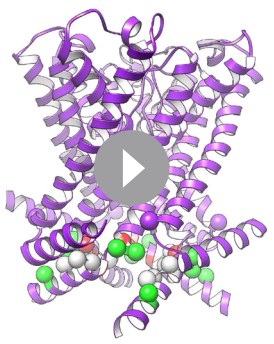

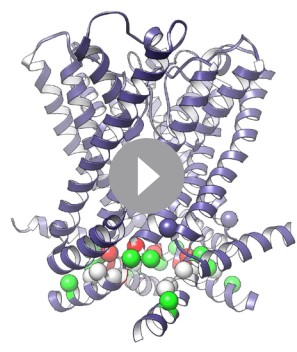

**Video 3.** Structural position of L291 in ABMD1 open model relative to previously probed positions in the C-linker. ABMD1 open state model highlighting positions of cysteine mutations from *Kwan et al., 2012* highlighted with their beta carbon as spheres. L291 is colored in the same color as the model while the positions on the C-linker are colored according to their functional effect with applied cadmium. Green indicates lock-open effect, red indicates lock-closed effect, and gray indicates no effect. Only the pore domain and A' helix of the C-linker are shown for clarity.
https://elifesciences.org/articles/80303/figures#video3

**Video 4.** Structural position of L291 in ABMD2 open model relative to previously probed positions in the C-linker. ABMD2 open state model highlighting positions of cysteine mutations from *Kwan et al., 2012* highlighted with their beta carbon as spheres. L291 is colored in the same color as the model while the positions on the C-linker are colored according to their functional effect with applied cadmium. Green indicates lock-open effect, red indicates lock-closed effect, and gray indicates no effect. Only the pore domain and A' helix of the C-linker are shown for clarity.
https://elifesciences.org/articles/80303/figures#video4

of a leucine zipper motif between S5 and S6, which demonstrated that disruption of this motif reduces the ability of the VSD to drive pore opening (*Wemhöner et al., 2012*). While stabilization of the S5/S6 interface upon opening is seemingly at odds with the model of Flynn and Zagotta, this difference can be reconciled by examining the mechanism of S6 gating. Voltage sensor activation displaces the lower S4/S5 interface away from the closed conformation of S6, enabling dilation of the

S6 helices. Thus, the resting conformation of S5 sterically hinders this outward and rotating movement of S6, in line with previous studies (*Flynn and Zagotta, 2018*).

Past studies on the role of the C-terminus in voltage gating of the channel have yielded mixed results. Truncation studies show the C-linker and CNBD are not necessary for channel activation, but mutagenesis studies have highlighted numerous interactions between the S4-S5 linker and C-terminus critical for electromechanical coupling (*Wainger et al., 2001*; *Decher et al., 2004*; *Kwan et al., 2012*). While our model for electromechanical coupling is primarily driven by interactions in the TMD, this model also involves coupling interactions between the S4-S5 linker and C-linker. The Yellen group has used cysteine crosslinking to extensively probe state-dependent interactions between the S4-S5 and C-linkers (*Kwan et al., 2012*). Out of the 23 pairs of residues probed in this study, 8 pairs favor the open state, 9 pairs favor the closed state, and the remaining were unresponsive to cadmium. Highlighting these interaction pairs by their functional

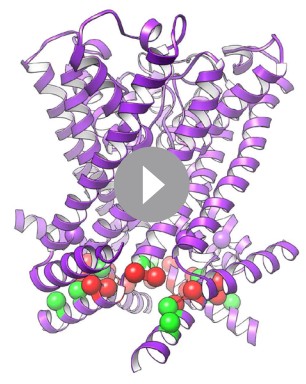

**Video 5.** Structural position of A294 in ABMD1 open model relative to previously probed positions in the C-linker. ABMD1 open state model highlighting positions of cysteine mutations from *Kwan et al., 2012* highlighted with their beta carbon as spheres. A294 is colored in the same color as the model while the positions on the C-linker are colored according to their functional effect with applied cadmium. Green indicates lock-open effect, red indicates lock-closed effect, and gray indicates no effect. Only the pore domain and A' helix of the C-linker are shown for clarity.
https://elifesciences.org/articles/80303/figures#video5

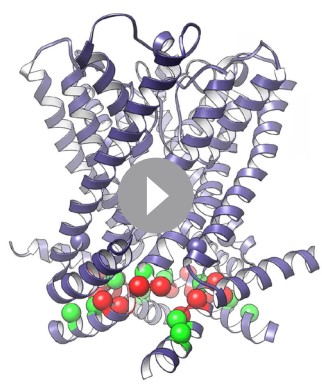

**Video 6.** Structural position of A294 in ABMD2 open model relative to previously probed positions in the C-linker. ABMD2 open state model highlighting positions of cysteine mutations from *Kwan et al., 2012* highlighted with their beta carbon as spheres. A294 is colored in the same color as the model while the positions on the C-linker are colored according to their functional effect with applied cadmium. Green indicates lock-open effect, red indicates lock-closed effect, and gray indicates no effect. Only the pore domain and A' helix of the C-linker are shown for clarity.
https://elifesciences.org/articles/80303/figures#video6

effect on channel gating in our ABMD open state models shows excellent overall agreement. The closest pairs of residues in our model correspond to the pairs identified to favor the open state while the more distant pairs favor the closed state or are not responsive to cadmium (*Videos 1–6*).

The open state model resulting from our ABMD simulation protocol involves a dilation at position 390 coupled to a rotation of V390 away from the central pore axis, resulting from bending of S6 at the upper glycine G382. The degree of rotation is low enough that V390 still lines the inner vestibule in the open state, consistent with the observation that this position influences the apparent affinity of open pore blockers (*Cheng et al., 2007*). Pore opening is coupled to deformation of the S6 helix and the formation of a π helix in several subunits. Interestingly, hERG is the only channel of the CNBD family for which the transition to π helix occurs in S6. Other CNBD family members, such as CNG channels, TAX-4, and HCN4, open via a more canonical right-handed twist of S6 at glycine 399 (*Zheng et al., 2020*; *Xue et al., 2021*; *Xue et al., 2022*). HCN1 may thus feature an opening mechanism that combines features of the CNBD family and those specific to hERG opening. Notably, in the closest homolog of resolved open structure, HCN4, the equivalent of V390 (I511), is rotated slightly but does not induce formation of a π-helix (*Figure 4—figure supplement 1*). Although it is possible that the difference observed between the HCN4 cryoEM structure and our simulations of HCN1 represents differences between isoforms, we suggest that this difference may also stem from the different functional states these models represent. The HCN4 structure was solved in the resting VSD conformation, thus corresponding to the resting-open state or the 'instantaneous' open state of HCN channels. The instantaneous open state is pharmacologically distinguishable from the activated-open state suggesting structural differences in the pore (*Proenza et al., 2002*).

The prevalence of state-dependent lipid interactions favoring the activated VSD and open pore highlights a potential role in electromechanical coupling. Saponaro et al. also report state-dependent lipid binding at the S4/S5/S6 interface in their closed and open structures of HCN4, though they observe lipid density only in the resting-closed state and not in the resting-open state (*Saponaro et al., 2021*). A similar lipid density was observed in the resting-closed structure of the hyperpolarization-activated plant channel KAT1 (*Clark et al., 2020*). Strikingly, a recent study reported recording from the related bacterial channel SthK in bilayers of controlled lipid composition and proposed a mechanism for the regulation of this channel by PA lipids (*Schmidpeter et al., 2022*). Indeed, cryoEM structures of this channel in different states where individual lipid molecules were resolved and showed lipid binding to a site containing residues corresponding to R297 and D401 engaged in a salt bridge interaction in the closed state of SthK (HCN1 numbering). This lipid is absent in the open state, and the aforementioned salt bridge is also broken, leading the authors to propose that PA acts in an activatory manner via an unlocking mechanism by binding in the closed state. Mutagenesis of the equivalent positions in HCN2 confirmed that this mechanism is likely conserved in HCN. Consistent with this model, our HCN1 opening simulations reveal reduced contacts between R297 and D401 and lipid headgroups in the activated/closed and in the activated/open state compared to the resting/closed state (*Figure 5*). Note, however, that our simulations, conducted in a simple model bilayer of POPC, do not permit to infer the role anionic lipids may play in regulating ion channel gating. Examining the state-dependence of lipid binding at the VSD-pore interface in more complex environments will be crucial for improving our understanding of lipid regulation of HCN channel function.

## Materials and methods

### Protein models

CryoEM structures of the resting (5U6P), intermediate (mutant Y289D, 6UQG), and the activated (6UQF) states of HCN1 were used as initial snapshots for MD simulations (*Lee and MacKinnon, 2019*; *Lee and MacKinnon, 2017*). Cyclic nucleotides solved in these structures were kept within each CNBD. A WT-model without the HCN domain (without residues 94–139) was also built using the HCN1 resting state (5U6P) as a starting model.

### System building

The different channel models were prepared using the CHARMM-GUI server (*Jo et al., 2008*; *Lee et al., 2016*). Each channel was embedded in a 1-palmytoyl-2-oleoyl-phosphatidyl- choline (POPC) bilayer (150 × 150 Å) and solvated in a 150 mM KCl solution. The CHARMM36 force field was used for the protein, lipids, and ions, and the TIP3P model for water (*Klauda et al., 2010*; *Huang and MacKerell, 2013*). The CgenFF module was used to parameterize cAMP molecules solved in HCN1 structures (*Vanommeslaeghe et al., 2010*).

### Molecular dynamics simulations

Minimization, equilibration, and production steps were performed on the Beskow-SNIC/Piz-daint supercomputers using, respectively, Gromacs 2019.3 and 2020.3 (*Abraham et al., 2015*; *Lindahl and Hess, 2021*). The standard CHARMM-GUI inputs were used for the minimization and equilibration of the systems. During these steps, harmonic restraints were applied to the protein-heavy atoms and the lipid heads and were gradually released during 1.2 ns. The production dynamics was then performed in the NPT ensemble without any restraints. Hyperpolarized or depolarized conditions were applied by imposing, respectively, a positive or negative electric field in the form of an external force acting on all charged particles of the system. Nose–Hoover thermostat (*Nosé, 1984*) and Parrinello–Rahman barostat (*Parrinello and Rahman, 1981*) were used to keep the temperature and the pressure constant at 310 K and 1 bar. Periodic boundary conditions were used and the particle mesh Ewald algorithm (*Darden et al., 1993*) was applied to treat long-range electrostatic interactions. A switching function was applied between 10 and 12 Å for the non-bonded interactions. LINCS (*Hess et al., 1997*) was applied to constrain the bond lengths involving hydrogen atoms. The integration timestep was set to 2 fs, and the overall lengths of the trajectories was 1 μs.

Longer simulations of equilibrium resting/closed HCN1 were performed using the Anton2 super-computer *Shaw et al., 2014*. These simulations were performed using standard CHARMM36 parameters in the NVT ensemble to avoid altering the applied electric field. The temperature was kept at 310 K using the Nose–Hoover thermostat. The multigrator approach was used for temperature and semi-isotropic pressure coupling. Long-range electrostatic interactions were handled using the

**Table 1.** Table summary of the molecular dynamics (MD) simulations carried out in this work. CryoEM structures (first column, PDB code in brackets) were simulated under different conditions (second column, including the presence/absence of specific domains/binding partners, under external electric field [EF] or not). The total simulation time is provided in the third column.

| Structure | Conditions | Time (ns) |
|---|---|---|
| HCN1 resting with cAMP (5U6P) | No EF | 1000 |
| HCN1 intermediate with cAMP (6UQG) | No EF | 1000 |
| HCN1 activated closed with cAMP (6UQF) | No EF | 1000 |
| HCN1 resting with cAMP (5U6P) | Without HCN domain + EF (–1 V) | 750 |
| HCN1 resting with cAMP (5U6P) | EF (–1 V) | 1000 |
| HCN1 resting with cAMP (5U6P) | EF (–550 mV) | 12,000 |
| HCN1 resting with cAMP (5U6P) | ABMD + EF (-) | 2 × 1000 |
| HCN1 activated with cAMP (6UQF) | ABMD + EF (-) | 2 × 1000 |

ABMD, adiabatic bias molecular dynamics.

u-series algorithm implemented in Anton2 (*Lippert et al., 2013*). The timestep was kept to 2 fs. All simulated structures and simulation details are listed in *Table 1*.

## Enhanced sampling MD simulations with ABMD

ABMD simulations were carried out using gromacs and the same simulation conditions described above (*Marchi and Ballone, 1999*). In this method, a ratchet-like biasing harmonic potential is added, encouraging further exploration of the CV space. In these simulations, Plumed2.5 (*Tribello et al., 2014*) was called along the MD production to apply a bias on three CVs, namely the distance between the center of mass of W281-N300, D290-K412, and pairs of opposite and adjacent V390. The bias was applied based on target distances that served as a threshold, where the bias between pairs was turned off if the distance evolved below (for W281-N300 and D290-K412) or above (V390) the target values. The distance target T0 was set to 6.0 Å for the W281-N300 and D290-K412 pairs and 11 Å and 16 Å for adjacent and opposite V390 pairs, respectively; and the bias force constant KAPPA equal to 4000.0 kJ/mol for each CV.

## MD simulations analysis

The contacts between helices were analyzed by Voronoi Laguerre Delaunay for Macromolecules (VLDM) (*Esque et al., 2010*). VLDM relies on a tessellation method, that is, a partition of space into a collection of polyhedra filling space without overlaps or gaps. Delaunay tessellation and its Laguerre dual were performed using a set of heavy-atom Cartesian coordinates and a weight that depends on the van der Waals radius of the atom, determined using the CHARMM36 force field. A contact occurs whenever two atoms share a common face in the tessellation. The interface between molecular groups is quantified by their polygonal surface area. The type of contacts in the interface determines the nature of the interface: hydrophobic contacts correspond to contacts between carbon atoms exclusively while electrostatic contacts involve N, O, S, and P atoms; hydrogen bonds and salt bridges both belong to this category.

## Visualization and analysis

Positions and distances were calculated using in-house tcl scripts for VMD. The S4 bending angle, which corresponds to the angle formed between the axis of upper S4 (254-272) and that of lower S4 (272-289), was calculated along MD trajectories using PLUMED v2.5. MD simulation trajectories were visualized using VMD (*Humphrey et al., 1996*) and PyMOL. Positions, distances, probability densities, angles, and contacts graphs were all generated using R v3.6.1 or gnuplot v5.0.

## Cell culture and transient expression

293T (ATCC, CRL-3216) were cultured in DMEM (Gibco, 11965-084) supplemented with 10% FBS (Gibco, 26140-079) and 2 mM L-glutamine (Corning, 25005CI). The cells were kept at 37°C and 8% $CO_2$ in a humidified incubator and were split every 3 d. For transfections, the cells were plated in a 12.5 $cm^2$ dish and were transfected at 60% confluency with 0.8 µg DNA using TransIT-293 (Mirus, MIR2700). We used a GFP-tagged and C-terminally truncated human HCN1 construct which was cloned into a pEG plasmid as previously reported by *Lee and MacKinnon, 2017*. Herein, we refer to this construct as HCN1-EM. For the double cysteine mutant, residues D290 and K412 were substituted with a cysteine each. The sequence of the construct was confirmed by DNA-sequencing. HCN1-EM or D290C-K412C channels were expressed in 293T cells for 24 hr and were plated at low density onto poly-L-lysine-coated 35 mm dishes before electrophysiological evaluation.

## Electrophysiology and data analysis

Macroscopic currents were recorded in whole-cell voltage-clamp configuration using an Axopatch 200b amplifier (Molecular Devices, USA). Signals were lowpass filtered at 2 kHz by a 4-pole Bessel filter and acquired at a sampling rate of 10 kHz using a Digidata 1440A (Molecular Devices). Data analysis was carried out in Clampfit 10 (Molecular Devices) and Origin (Origin Lab Corporation, USA). The bath solution contained 130 mM NaCl, 10 mM KCl, 1.8 mM $CaCl_2$, 0.5 mM $MgCl_2$, 5 mM HEPES, pH 7.4 NaOH. The pipette resistance was in the range of 1–2 MΩ, using a pipette solution consisting of 130 mM KCl, 10 mM NaCl, 0.5 mM $MgCl_2$, 2 mM Mg-ATP, and 5 mM HEPES, pH 7.2 KOH. For the cysteine-crosslinking experiment, we added 100 µM cadmium to the pipette solution. To allow for

equilibration of the cytosol with the pipette solution, we waited at least 2 min after breaking into the cells.

HCN1-EM-expressing cells were clamped to a holding potential of –30 mV. Substitution of residues D290 and K412 with cysteins resulted in a shift of the half-maximal activation to depolarized potentials. The cells were therefore clamped to a holding potential of +60 mV, a potential at which the channels were in the resting state. To assess the voltage-dependence of activation, we used a two-pulse protocol in which the channels were preconditioned to voltages in the range of +60 mV (HCN1-D290C-K412C) to –140 mV with –20 mV increments, before tail currents at a test pulse of –140 mV, a potential of maximal channel activation, were recorded. The inter-sweep time was set to 30 s to ensure complete deactivation before the next pre-condition pulse was applied. Tail currents recorded herein are transient currents, where the initial current resembles an ensemble current of channels activated during the pre-conditioning pulse, and which increases during the test pulse until the maximal open probability is reached. Thus, the initial current of the tail current plotted as a function of voltage can be used to obtain channel activation. The activation curve was normalized and fitted with a single Boltzmann function: $y = (A1 – A0)/(1 + \exp((V – V0.5)/k)) + A0$, where $A0$ and $A1$ are the minimum and maximum tail current, $V0.5$ is the potential of half maximal channel activation, $V$ is the membrane potential, and $k$ is the slope factor. In HCN1-D290C-K412C-expressing cells, we observed a high basal activity. To assess whether the basal activity was a result of leak current or due to increased channel activity, we performed cesium-blocking experiments, in which we perfused the cells with 1 mM cesium-containing extracellular solution. Cesium is a potassium channel blocker and inhibits HCN1-mediated currents too. Thus, the remaining current seen during cesium application can be used to assess the leak current, which occurs because of poor seal resistance or through cesium-resistant channels. Data are given as mean ± SEM for the indicated number of experiments (n).

## Acknowledgements

This work was funded by the Science for Life Laboratory (LD), the Göran Gustafsson Foundation (LD), the Swedish Research Council VR 2018-04905, 2019-02433, and 2022-04305 to LD, funding from NINDS (1R35NS116850 to BC) and the Austrian Science Foundation's Schroedinger (J4652) fellowship (VB). The MD simulations were performed on resources provided by the Swedish National Infrastructure for Computing (SNIC) on Beskow at the PDC Center for High Performance Computing (PDC-HPC).

## Additional information

### Competing interests

Lucie Delemotte: Reviewing editor, *eLife*. The other authors declare that no competing interests exist.

### Funding

| Funder | Grant reference number | Author |
|---|---|---|
| Science for Life Laboratory | | Lucie Delemotte |
| Gustafsson foundation | | Lucie Delemotte |
| Vetenskapsrådet | 2018-04905 | Lucie Delemotte |
| Vetenskapsrådet | 2019-02433 | Lucie Delemotte |
| Vetenskapsrådet | 2022-04305 | Lucie Delemotte |
| National Institute of Neurological Disorders and Stroke | 1R35NS116850 | Baron Chanda |
| Austrian Science Foundation | J4652 | Verena Burtscher |

The funders had no role in study design, data collection and interpretation, or the decision to submit the work for publication.

## Author contributions
Ahmad Elbahnsi, Conceptualization, Formal analysis, Supervision, Validation, Investigation, Visualization, Writing - original draft; John Cowgill, Conceptualization, Formal analysis, Investigation, Visualization, Writing - original draft, Writing - review and editing; Verena Burtscher, Formal analysis, Investigation, Writing - original draft; Linda Wedemann, Luise Zeckey, Data curation, Investigation, Visualization; Baron Chanda, Conceptualization, Supervision, Funding acquisition, Investigation, Project administration, Writing - review and editing; Lucie Delemotte, Conceptualization, Resources, Supervision, Funding acquisition, Investigation, Visualization, Project administration, Writing - review and editing

## Author ORCIDs
Ahmad Elbahnsi (ID) http://orcid.org/0000-0002-5356-2440
Verena Burtscher (ID) http://orcid.org/0000-0002-0464-0799
Linda Wedemann (ID) http://orcid.org/0000-0003-1096-9037
Baron Chanda (ID) http://orcid.org/0000-0003-4954-7034
Lucie Delemotte (ID) http://orcid.org/0000-0002-0828-3899

## Decision letter and Author response
Decision letter https://doi.org/10.7554/eLife.80303.sa1
Author response https://doi.org/10.7554/eLife.80303.sa2

## Additional files

### Supplementary files
• MDAR checklist

### Data availability
The files containing the raw data for the modeling/computational part of the study can be found at the following link: https://zenodo.org/record/7920679#.ZFyJW-zRY-S The scripts used to analyze the MD simulations can be found at https://github.com/elbahnsi/ELIFE-Interplay-between-VSD-pore-and-membrane-lipids-in-electromechanical-coupling-in-HCN-channels, (copy archived at swh:1:rev:50e5505b8d4300fa9163155b46d0b0fa8dbd7440). All data generated or analyzed during the experimental part of the study are included in the manuscript.

The following dataset was generated:

| Author(s) | Year | Dataset title | Dataset URL | Database and Identifier |
| --- | --- | --- | --- | --- |
| Ahmad E | 2023 | Interplay between VSD, pore and membrane lipids in electromechanical coupling in HCN channels | https://zenodo.org/record/7920679#.ZFyJW-zRY-S | Zenodo, 10.5281/zenodo.7920679 |

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
