## [Editor Report]

In this study the authors aim to describe the electromechanical coupling responsible for activation of a Hyperpolarised-activated and Cyclic Nucleotide-gated (HCN) channel. HCN channels are the only mammalian channels to open under hyperpolarisation, being important for their roles in cardiac and neuronal cells. The authors use enhanced-sampling atomistic simulations to enforce sampling between open and closed states of the channel. The simulations suggest state-dependent interactions involving pore and voltage sensor helices, as well as with lipids, leading the authors to propose a domino-like mechanism of activation. These findings will be of considerable interest to the ion channel community.

---

## [Decision Letter]

**Decision letter after peer review:**

Thank you for submitting your article "Interplay between VSD, pore and membrane lipids in electromechanical coupling in HCN channels" for consideration by *eLife*. Your article has been reviewed by 3 peer reviewers, one of whom is a member of our Board of Reviewing Editors, and the evaluation has been overseen by Richard Aldrich as the Senior Editor. The reviewers have opted to remain anonymous.

Essential revisions:

The reviewers were supportive of the work and appreciated the mechanistic insights. The main concerns needing to be addressed are listed below, with full details available in the reviews themselves.

1. The decision to model the open pore of HCN1 using homology with hERG has been questioned. Because the simulations depend critically on this open-state model, it requires validation. It has been suggested that a better choice would have been the available HCN4 open state cryoEM structure. It has also been stated that hERG could be problematic because it does not have a gating hinge in common with HCN and has other sequence differences (such as V390) that may impact the structure of the HCN1 model. The authors should try the HCN4 pore domain to see how such a homology model would compare to the one used.

2. Provide experimental evidence in the form of existing or new experimental data to demonstrate the accuracy of the current open-state HCN1 model that has been used to set the interaction distances to guide simulations. In particular, data is requested that can confirm the relevance of those interaction distances. Also, for D290 – K412, the validation is missing a control (effect of Cd2+ on wildtype).

3. Provide any available mutagenesis or other data that supports the importance of residues proposed to interact with lipids.

4. Provide analysis or additional simulation that can address concerns about the reproducibility of simulation results, given only 1 simulation that enforces the chosen simulation distances has been performed. Also, please provide statistical tests to demonstrate an increase vs decrease in key interactions.

5. Better explain and visualise the proposed mechanism referred to as a "domino effect".

*Reviewer #1 (Recommendations for the authors):*

I note that the low-resolution blurry pdf figures provided for review made the reading of details very difficult.

I must say the figures have not done a good job of explaining the so-called "domino" effect, and perhaps data can be presented alongside a cartoon to explain this, with the domino nature clearly explained, just as the "domino effect" has been explained clearly in pentameric ligand-gated channels, for example.

Additional comment/question regarding the D290-K412 salt bridge cysteine cross-bridging: Is it possible that this approach might capture rare conformations, where those cysteines may briefly come to within 8 Å and then bind for a long time, despite not being relevant to the wildtype functional state? Irreversible binding (on the experimental timescale) may trap unphysical conformations, reminiscent of what has been previously argued in relation to biotin-avidin experiments examining VSD movements (e.g. Jogini and Roux Biophys J 2007 93:3070), for example. It is not clear to me that the cross-link proves that a wildtype salt bridge is important for the activated state, and why in the main review I ask about possible other experiments to back it up.

Figure 3c shows some data for D290-K412. I note that supporting timeseries and violin plots like Figure 3C would be useful for all relevant distances, including N300-W281, V390-I302 and others noted.

*Reviewer #2 (Recommendations for the authors):*

This is a well-written manuscript on a hot topic. The study would attract many readers. But the figures are extremely unclear, the colors are not helpful, and the text in the figures is too small. There are also some other concerns that need to be addressed.

1. It is not clear why W281 was indicated as a very interesting residue. Is there something different about this residue compared to other residues in the channel? It seems to come out of the blue that this residue is especially interesting to study. Or figure 2 is maybe the true reason of the focus?

2. Figure 1. W281 interactions with 277-286: are these interactions intersubunit or intrasubunit? If intrasubunit, is it really interesting if neighboring sidechains on an α helix are interacting? Do they stabilize some conformation?

3. Pg. 6. "the Y289D mutant retains many characteristics of the VSD-pore interactions of the activated state". At this point you have only stated that W281 only interacts with lipids in the activated state, so what VSD-pore interactions are you referring to here?

4. "The wider distribution of bending angles in S4 for the system lacking the HCN domain indicates that the HCN domain may also contribute to stabilizing the activated state of S4 (Figure 2 —figure supplement 1B and 2F)." This is not clear in Figure 2 supp 1B or 2F.

5. Do you have any experimental data to support your proposed stabilization of the open state by the W281-N300 interaction? Or could it be validated in the MD by mutating them?

6. Figure 2 Suppl 2. Do you have any statistical test for the increase vs decrease of the interactions? Not clear to me of any consistent changes in some cases.

7. Pg. 15. "However, this distance decreases rapidly in the first few picoseconds of simulations" What simulation? Starting from what state?

8. Figure 3 Suppl 1. The channels should really be more closed by the C-C mutation if D-K stabilizes the open state, i.e. the voltage dependence should be shifted to more negative voltages and channels should be less open. Right now, the cys-cys mutation does the opposite, shifts the voltage dependence to more depolarized voltages and the channels are less closed (more leak currents). So, it is not really clear that a D-K interaction really stabilizes the open state. Could it be that cyc-cys +Cd2+ stabilizes the open state (because these residues are close in space), but D-K does not stabilize the open state even if they are nearby each other?

9. Figure 4 Suppl 3. I cannot see the proposed increases and decreases in S6-S5 and S6-S6 interactions, respectively. Is there some statistics or number to bolster these claims? Should not the ABMD curve go from resting towards the open state during the simulations? This is not always the case?

10. Figure 5A. I cannot see the proposed increase in lipid contacts on S5 and S6? Can you quantify something? Why not use the ABMD state instead of the activated state, since the activated state is not open?

11. The Domino effect is not very well explained. What is happening in the different steps during the domino effect, i.e. what step is first and why does this step cause the second step, etc…? Figure 6 is not clearly showing in a figure what you think is happening. Also, some residues and symbols are shown without any explanations: what do the arrows under "int" represent, and what are F and E?

12. The role of lipids in hyperpolarization-dependent gating is not well documented or described. How is this supposed to happen and how can it be tested? Not clear what H392-R297 and their lipid interaction are doing?

[Editors' note: further revisions were suggested prior to acceptance, as described below.]

Thank you for resubmitting your work entitled "Interplay between VSD, pore and membrane lipids in electromechanical coupling in HCN channels" for further consideration by *eLife*. Your revised article has been evaluated by Kenton Swartz (Senior Editor) and a Reviewing Editor.

We do note, however, that the revised manuscript was seen by the past reviewers and they strongly recommend making a couple of changes summarised below.

Recommended changes:

1. The authors refer to the effect of a lipid on the salt bridge R297 and D401, recently identified in HCN2, as a general feature of all HCN (Schmidpeter et al. 2022. Nat. Struct. Mol. Biol. 29, 1092-11009). This evidence is questionable, as the salt bridge is not present and plays no role in other HCN subtypes, for instance in HCN4. Examination of the 4 structures of HCN in the closed state in databank, 2 from hHCN1 and 2 from hHCN4, there is no clear evidence of the salt bridge between R297 and D401 (HCN2 numbering). First of all, there is NO density for the side chain of D, so it cannot be modelled. Note that the absence of density is by itself a strong indication that it does not form a salt bridge. Aspartates are subjected to radiation damage and it is known that when their sidechains form a salt bridge they are preserved, otherwise they are damaged by the electrons during image collections. If, nonetheless, one wanted to model the side chain of D, in any case, the distance from R side chain is in all 4 cases >4 A, above the cut-off distance for a salt bridge. It is therefore recommended that the authors add a statement admitting the lack of evidence for the conservation of a salt bridge pair in HCN1 and HCN4.

2. Dai and Zagotta 2017 do not say anything about HCN channels, so this reference should be removed in most places.

3. Page 22. Yellen did not show H462-Q468C, but H462-L466C Cd2+ crosslinking. So this data does not support the bond equivalent to H392-Q398 that the authors propose.

---

## [Author Response]

Essential revisions:The reviewers were supportive of the work and appreciated the mechanistic insights. The main concerns needing to be addressed are listed below, with full details available in the reviews themselves.1. The decision to model the open pore of HCN1 using homology with hERG has been questioned. Because the simulations depend critically on this open-state model, it requires validation. It has been suggested that a better choice would have been the available HCN4 open state cryoEM structure. It has also been stated that hERG could be problematic because it does not have a gating hinge in common with HCN and has other sequence differences (such as V390) that may impact the structure of the HCN1 model. The authors should try the HCN4 pore domain to see how such a homology model would compare to the one used.

We thank the reviewers for pointing this out. We concur that the choice of target model largely guides the outcomes of the biased simulations, and have reconsidered our choices in the light of the points raised, in particular by reviewer 3.

To reduce the impact of the template choice, we have decided to remove all the results that were based on homology modeling, and have decided instead to run several replicas of biased simulations where a distance that consistently changes upon opening in CNBD channels of known closed and open structures is used as a collective variable. Indeed, there are subtle differences between the CNG open, hERG open and HCN4 open state structures, but they all have in common a widening of the pore at the position equivalent to V390.

This has thus led us to run ABMD simulations biasing, on top of the D290-K412 and W281N300 distance pairs which were used in the original preprint, distances between V390 of neighboring and opposite subunits (Figure 4A). We thus added a ratchet bias that encourages V390 pairs to be pushed away from one another. This choice is based on the fact that residues at positions equivalent to V390 in other CNBD channels consistently show distance increases associated with dilation of inner pore gates (in TAX-4, this distance increases from 10.6 to 17.2 Å (PDB IDs 6WEJ-6WEK), in CNGA1 from 9.6 to 16.4 Å (PDB IDs 7LFT-7LFW), in CNGA1/B1 from 9.8 to 15.1 Å (PDB IDs 7RH9-7RHH), in EAG/hERG from 10.7 to 17.3 Å (PDB IDs 5KL7L5VA1) and in HCN4 from 10.5 to 15.0 Å (PDB IDs 7NP4-7NP3)). reminiscent of what happens in hERG is also at play here (Figure 4 S1).

We also noticed that the simulations initiated in the resting state did not fully activate (VSDs remind partially deactivated, likely because of the lack of collective variables explicitly encouraging VSD activation) and we thus focused the rest of the analysis of the electromechanical coupling mechanism on the simulations initiated in the activated state (Figure 4C-F, Figure 4 S2, Figure 5).

Some details of the coupling mechanism and the lipid-protein interactions change with this new set of simulations, but the coarse features of the domino effect previously described hold with these new CVs. We have revised the aforementioned figures, and their description in the text, as well as the methods section accordingly.

2. Provide experimental evidence in the form of existing or new experimental data to demonstrate the accuracy of the current open-state HCN1 model that has been used to set the interaction distances to guide simulations. In particular, data is requested that can confirm the relevance of those interaction distances. Also, for D290 – K412, the validation is missing a control (effect of Cd2+ on wildtype).

As discussed above, we agree with the reviewers that support for the V390-I302 interaction was lacking and instead used a distance compatible with all observed open states in the CNBD family. The W281-N300 interaction in the open state is supported by mutagenesis of these positions in spHCN by the Larsson group (Ramentol et al., 2020; Wu et al., 2021) as reviewer 3 has pointed out. Our experiments in Figure 3 and Figure 3-supplement 1 as the works of Gary Yellen support the D290-K412 interaction upon opening. We also added the requested control experiment to show that intracellular cadmium causes minimal effects to gating in the HCN1 wildtype construct. We have updated our text to more clearly highlight the past works in motivating us to use these interactions in our ABMD simulations.

As for validating our open state model from ABMD, we have followed the suggestions of reviewer 2 and compared our open state models with the cross bridging mutations on the S4-S5 linker and C-linker of spHCN from the Yellen group (Kwan et al. 2012). Our model agrees well with the functional effects observed in that study, with the residues on the C-linker nearest to the probed site on the S4-S5 linker in our model corresponding to positions shown to favor the open state upon cadmium bridging. We have added new videos (Videos 1-6) to map these 23 probed interactions onto our two open state models obtained from replicate ABMD runs.

Additionally, the increased S5-S6 interfacial contact upon pore opening that we observe in ABMD and propose as a central component of electromechanical coupling is supported by mutagenesis of the leucine zipper motif between S5 and S6 (Wemhöner, et al. 2012). Disruption of this leucine zipper motif with alanine mutants weakens VSD-pore coupling, supporting our domino effect model where the increased hydrophobic interactions between S5 and S6 compensate for the weakened S6-S6 interface. We have added this reference into our updated Discussion section.

3. Provide any available mutagenesis or other data that supports the importance of residues proposed to interact with lipids.

It is notoriously difficult to obtain experimental evidence of the implication of specific residues in lipid/protein interactions.

At the time we submitted our preprint, it was well established that HCN channel function is potentiated by the signaling lipids PIP2 and PA (Zolles et al. Neuron 52, 1027–1036 (2006) ; Fogle et al. J. Neurosci. 27, 2802–2814 (2007) ; Pian et al. J. Gen. Physiol. 128, 593–604 (2006)) and that direct lipid binding can be visualized in cryoEM structures (Saponaro et al. Mol. Cell 81, 2929–2943.e6 (2021) ; Lee and MacKinnon, Cell 179, 1582–1589.e7 (2019)). However, the functional impact of these binding events remain to be fully understood.

The most direct evidence for functionally important lipid-protein interactions requires being able to carry out mutagenesis experiments while controlling the lipid environment in electrophysiology recordings, which has so far never been reported for HCN channels. While our paper was under revisions, however, an important study was published by the Nimigean lab (Schmidpeter et al. 2022. Nat. Struct. Mol. Biol. 29, 1092–1100), which reported the successful recordings from the simulations were all conducted in POPC, and establishing a role for other (anionic) lipids, awaits further work.

These considerations are now mentioned in the main text.

4. Provide analysis or additional simulation that can address concerns about the reproducibility of simulation results, given only 1 simulation that enforces the chosen simulation distances has been performed. Also, please provide statistical tests to demonstrate an increase vs decrease in key interactions.

We thank the reviewers for this comment. Indeed, we agree that our statistical analysis was not satisfactory. As mentioned in the response to essential revisions point 1, we have now carried out replicates of our ABMD simulations and the analyses we report are either based on four replicates (opening and conduction) or on two replicates (analysis of the coupling mechanism), revealing consistency of the proposed mechanism across replicates.

In addition, to address the statistical test request, we have chosen to report directly probability distributions where appropriate, and include an estimate of the standard deviation for our bar charts reporting interactions or contacts between residues or residues and lipids and the interaction surface areas. Performing statistical tests on MD simulations data is non-trivial because the simulation frames are correlated along time and establishing firmly what decorrelation time is needed to obtain independent samples is far from straightforward. We thus reason that showing the data in a way where the reader can form their own opinion about the overlap in distributions is the most honest representation of the data.

5. Better explain and visualise the proposed mechanism referred to as a "domino effect".

We thank the editor and reviewers for the request. We have revised the figure to describe better the different steps of the “domino-like” mechanism and the corresponding text in the manuscript. We believe this was indeed an essential revision in terms of conveying our mechanistic insights.

Reviewer #1 (Recommendations for the authors):I note that the low-resolution blurry pdf figures provided for review made the reading of details very difficult.

We understand the concern and provide revised figures at higher resolution in the resubmitted paper.

I must say the figures have not done a good job of explaining the so-called "domino" effect, and perhaps data can be presented alongside a cartoon to explain this, with the domino nature clearly explained, just as the "domino effect" has been explained clearly in pentameric ligand-gated channels, for example.

Noted. Please see the response to essential revisions point 5.

Additional comment/question regarding the D290-K412 salt bridge cysteine cross-bridging: Is it possible that this approach might capture rare conformations, where those cysteines may briefly come to within 8 Å and then bind for a long time, despite not being relevant to the wildtype functional state? Irreversible binding (on the experimental timescale) may trap unphysical conformations, reminiscent of what has been previously argued in relation to biotin-avidin experiments examining VSD movements (e.g. Jogini and Roux Biophys J 2007 93:3070), for example. It is not clear to me that the cross-link proves that a wildtype salt bridge is important for the activated state, and why in the main review I ask about possible other experiments to back it up.

Cadmium crosslinking has been widely used throughout the ion channel field to probe the functional effects of putative interacting pairs. This has been especially true for the HCN field where work from the Yellen group has probed the structural changes in the VSD, C-linker, and gate using this approach. Furthermore, the activated state of the HCN1 VSD was obtained by the McKinnon group using cryoEM with mercury cross linking of S4 to S2 to stabilize the activated state in the absence of an electric field. The activated state structure obtained with this approach agreed well with our previous simulations of HCN1 VSD activation under an electric field.

Though it is still possible that this approach captures rare conformations, we feel two main lines of support the D290-K412 interaction as physiologically relevant. First, the salt bridge forms spontaneously in the simulations of the activated state (shown more clearly in our updated figure 3C) and was observed in the Y289D mutant structure. Next, the Yellen group has shown that cross bridging many sites in this region in spHCN stabilizes the open state. We have added videos comparing his results to our open state models and added a section to the discussion to highlight this comparison.

Figure 3c shows some data for D290-K412. I note that supporting timeseries and violin plots like Figure 3C would be useful for all relevant distances, including N300-W281, V390-I302 and others noted.

We have reported probability distributions for key interactions at the S4/S5 interface (W281N300 in Figure 1 S1) and important for pore opening (H392-Q398 and R404-E282 in Figure 4 S1).

Reviewer #2 (Recommendations for the authors):This is a well-written manuscript on a hot topic. The study would attract many readers. But the figures are extremely unclear, the colors are not helpful, and the text in the figures is too small. There are also some other concerns that need to be addressed.

1. It is not clear why W281 was indicated as a very interesting residue. Is there something different about this residue compared to other residues in the channel? It seems to come out of the blue that this residue is especially interesting to study. Or figure 2 is maybe the true reason of the focus?

The interactions between residues 277-286 and W281 are intrasubunit. We report them as a change in interaction pattern within a helix indicates the reorientation of the buly Trp sidechain, which is presumably important functionally.

The reason we have zoomed in onto this residue is three-fold:

The orientation of this residue is strikingly different in the three structures, as shown in figure 1. Given the bulkiness of this residue and its rarity, we reasoned it may play a significant functional role.Our preliminary simulations indicated that complete activation of the VSDs, as described in our previous work by a full transfer of the gating charge residues and a bending of the S4 helix at S272 (Kasimova et al. *eLife* 8:e53400 (2019)), is only fully achieved when W281 rotates and orients it side chain towards N300.Ramentol et al. (Ramentol et al. Nat Commun 11, 1419 (2020)) show that this residue pair plays a crucial role to determine the polarity of gating, indicating a functional significance, and prompting us to study more deeply the interaction patter of the S4 residue of the pair.

We have modified the text to explain our rationale in more detail.

2. Figure 1. W281 interactions with 277-286: are these interactions intersubunit or intrasubunit? If intrasubunit, is it really interesting if neighboring sidechains on an α helix are interacting? Do they stabilize some conformation?

The interactions between residues 277-286 and W281 are intrasubunit. We report them as a change in interaction pattern within a helix indicates the reorientation of the buly Trp sidechain, which is presumably important functionally.

3. Pg. 6. "the Y289D mutant retains many characteristics of the VSD-pore interactions of the activated state". At this point you have only stated that W281 only interacts with lipids in the activated state, so what VSD-pore interactions are you referring to here?

We are referring to the ensemble of interactions at the S1-S4-S5 interface described in the 3 paragraphs above.

4. "The wider distribution of bending angles in S4 for the system lacking the HCN domain indicates that the HCN domain may also contribute to stabilizing the activated state of S4 (Figure 2 —figure supplement 1B and 2F)." This is not clear in Figure 2 supp 1B or 2F.

We had miscited the figure. We have edited the text to reflect that there is a reduced correlation between S4 bending angle and W281 rotation in absence of HCN domain (as seen by comparing Figures 2C and 2F).

5. Do you have any experimental data to support your proposed stabilization of the open state by the W281-N300 interaction? Or could it be validated in the MD by mutating them?

Ramentol and Larsson (Ramentol et al. Nat Commun 11, 1419 2020) have shown that this residue pair plays a crucial role in determining the polarity of gating, indicating a functional significance of their interaction. Whether their interaction indeed stabilizes the open state awaits experimental validation, however.

6. Figure 2 Suppl 2. Do you have any statistical test for the increase vs decrease of the interactions? Not clear to me of any consistent changes in some cases.

Given that these simulations are not performed at equilibrium but under a transmembrane voltage bias, it is difficult to report probability distributions. As mentioned above, statistical testing is complicated in equilibrium simulations, making it even more complicated for nonequilibrium simulations. We have toned down our claims in the text describing this figure.

7. Pg. 15. "However, this distance decreases rapidly in the first few picoseconds of simulations" What simulation? Starting from what state?

We were implicitly referring to simulations of the activated VSD-closed pore structure, we have modified the text to make this explicit.

8. Figure 3 Suppl 1. The channels should really be more closed by the C-C mutation if D-K stabilizes the open state, i.e. the voltage dependence should be shifted to more negative voltages and channels should be less open. Right now, the cys-cys mutation does the opposite, shifts the voltage dependence to more depolarized voltages and the channels are less closed (more leak currents). So, it is not really clear that a D-K interaction really stabilizes the open state. Could it be that cyc-cys +Cd2+ stabilizes the open state (because these residues are close in space), but D-K does not stabilize the open state even if they are nearby each other?

We agree that the rightward shift and increased basal activity of the D290C-K412C mutant relative to wild-type mutant was unexpected given that this interaction should stabilize the open state. However, there are several possible explanations for this effect. First, some D290CK412C may spontaneously form disulfide bonds during the recording due to the absence of reducing agents in the internal pipette solution. This would be consistent with the further rightward shift and increased basal activity of the D290C-K412C mutant in the presence of cadmium. We waited 2+ minutes after breaking into the cell to initiate recordings, which lasted for several minutes so some degree of disulfide bond formation would not be unexpected. Additionally, D290 and K412 may be involved in closed-state-specific interactions that are disrupted by the cysteine mutations. We have updated our Results section to include discussion of these possibilities.

9. Figure 4 Suppl 3. I cannot see the proposed increases and decreases in S6-S5 and S6-S6 interactions, respectively. Is there some statistics or number to bolster these claims? Should not the ABMD curve go from resting towards the open state during the simulations? This is not always the case?

We have added bar plots comparing these interactions in the resting, activated, and ABMD simulations initiated in the activated state, and included standard deviations as error bars on these graphs. From the analysis of the new ABMD data (see response to essential revisions point 1), it emerges that the S6/S6 interface clearly decreases, while the change in the S5/S6 interface is less striking. We have updated the text to reflect this.

10. Figure 5A. I cannot see the proposed increase in lipid contacts on S5 and S6? Can you quantify something? Why not use the ABMD state instead of the activated state, since the activated state is not open?

We thank the reviewer for the good suggestion, we now present interaction plots (including uncertainties as standard deviations) containing data from the resting/closed, the activated/closed and the activated/open ABMD simulations. Correspondingly, in figure 5D we have also introduced a panel showing the molecular model in the activated/open state postABMD.

The most striking change in contact frequency when changing from resting/closed, to activated/closed and to activated/open states involve a decrease in contact frequency between POPC headgroups and Ile275, His279, Arg297, Asn300 and Asp 401 and a increase in contacts frequency with Trp281 and Leu 291.

We have modified figure 5 and amended the text to reflect this analysis.

11. The Domino effect is not very well explained. What is happening in the different steps during the domino effect, i.e. what step is first and why does this step cause the second step, etc…? Figure 6 is not clearly showing in a figure what you think is happening. Also, some residues and symbols are shown without any explanations: what do the arrows under "int" represent, and what are F and E?

We have modified the figure and the accompanying text, please refer to the answer to essential revisions point 5.

12. The role of lipids in hyperpolarization-dependent gating is not well documented or described. How is this supposed to happen and how can it be tested? Not clear what H392-R297 and their lipid interaction are doing?

We agree with the reviewer and think of our work as providing hypotheses that can be further probed. We were nevertheless happy to find a few experimental results in support of our hypothesis, as detailed in our response to essential revisions point 3.

[Editors' note: further revisions were suggested prior to acceptance, as described below.]

Recommended changes:1. The authors refer to the effect of a lipid on the salt bridge R297 and D401, recently identified in HCN2, as a general feature of all HCN (Schmidpeter et al. 2022. Nat. Struct. Mol. Biol. 29, 1092-11009). This evidence is questionable, as the salt bridge is not present and plays no role in other HCN subtypes, for instance in HCN4. Examination of the 4 structures of HCN in the closed state in databank, 2 from hHCN1 and 2 from hHCN4, there is no clear evidence of the salt bridge between R297 and D401 (HCN2 numbering). First of all, there is NO density for the side chain of D, so it cannot be modelled. Note that the absence of density is by itself a strong indication that it does not form a salt bridge. Aspartates are subjected to radiation damage and it is known that when their sidechains form a salt bridge they are preserved, otherwise they are damaged by the electrons during image collections. If, nonetheless, one wanted to model the side chain of D, in any case, the distance from R side chain is in all 4 cases >4 A, above the cut-off distance for a salt bridge. It is therefore recommended that the authors add a statement admitting the lack of evidence for the conservation of a salt bridge pair in HCN1 and HCN4.

This comment prompted us to closely examine the existing cryoEM structures. In the closed HCN1, (Author response image 1), it appears that the interaction between R297 and D401 is modeled as a salt bridge. It is true that the D401 density isn't great. In fact, the side chain density overall isn't great because the structure is relatively old at this point and data processing has improved greatly since then.

**Author response image 1. sa2fig1:** overlay between model (light green helices) and density (EMD-8512) for the closed state hHCN1 model (5U6P).

For HCN4 we have two closed-state structures in the PDB (Author response image 2 and Author response image 3). In both, the same interaction is modeled as a salt bridge. The side chain density is not unambiguous but a small bulge appears to be present in both cases.

**Author response image 2. sa2fig2:** Figure R2: overlay between model (light pink helices) and density (EMD-0093) for the closed state hHCN4 model (6GYN).

**Author response image 3. sa2fig3:** Figure R3: overlay between model (bright pink helices) and density (EMD-12513) for the closed state rabbit HCN4 model (7NP4).

Interestingly, in support of our hypothesis, the distance appears much larger in the rabbit HCN4 open state model (Author response image 4). This would yield a weaker interaction, leading to a model in which this interaction would stabilize the closed state.

**Author response image 4. sa2fig4:** Figure R4: overlay between model (light blue helices) and density (EMD-12512) for the open state rabbit HCN4 model (7NP3).

While we find the comment by the reviewer about the salt bridge protecting the side chain thought-provoking, we compared the density in the voltage-sensor domain, where strong salt bridges between residues of opposite charges are well established (Author response image 5).

**Author response image 5. sa2fig5:** Zoom in onto the VSD region of the closed state rabbit HCN4 model. Overlay between model (bright pink helices) and density highlighting a well-established salt bridge interaction.

In addition to the structural evidence above, mutagenesis in both spHCN and HCN2 supports the proposed role of this salt bridge in stabilizing the closed pore (Decher, *et al.* 2004, Flynn and Zagotta, 2018). Given that spHCN and HCN2 diverged much earlier than the mammalian subtypes, it is likely that the role of this salt bridge is conserved across HCN1-4.

Taken together, we believe the structural evidence for the relevance of the state-dependency of this interaction across channels from the HCN channel family is rather strong.

2. Dai and Zagotta 2017 do not say anything about HCN channels, so this reference should be removed in most places.

Thank you for pointing out this issue. This reference has been removed where appropriate.

3. Page 22. Yellen did not show H462-Q468C, but H462-L466C Cd2+ crosslinking. So this data does not support the bond equivalent to H392-Q398 that the authors propose.

Rothberg et al. 2003 do in fact probe the interaction between the equivalent of H462-Q468C in spHCN. The phenotype is however complex and the authors are thus quite tentative about calling it a lock-open effect. We have thus removed this reference from our manuscript.